# A versatile Halo- and SNAP-tagged BMP/TGFβ receptor library for quantification of cell surface ligand binding

Jerome Jatzlau [1,5], Wiktor Burdzinski [1,2,5], Michael Trumpp [1], Leon Obendorf[1], Kilian Roßmann[3], Katharina Ravn[4], Marko Hyvönen [4], Francesca Bottanelli[1], Johannes Broichhagen [3] & Petra Knaus [1,2✉]

TGFβs, BMPs and Activins regulate numerous developmental and homeostatic processes and signal through hetero-tetrameric receptor complexes composed of two types of serine/threonine kinase receptors. Each of the 33 different ligands possesses unique affinities towards specific receptor types. However, the lack of specific tools hampered simultaneous testing of ligand binding towards all BMP/TGFβ receptors. Here we present a N-terminally Halo- and SNAP-tagged TGFβ/BMP receptor library to visualize receptor complexes in dual color. In combination with fluorescently labeled ligands, we established a Ligand Surface Binding Assay (LSBA) for optical quantification of receptor-dependent ligand binding in a cellular context. We highlight that LSBA is generally applicable to test (i) binding of different ligands such as Activin A, TGFβ1 and BMP9, (ii) for mutant screens and (iii) evolutionary comparisons. This experimental set-up opens opportunities for visualizing ligand-receptor binding dynamics, essential to determine signaling specificity and is easily adaptable for other receptor signaling pathways.

[1] Institute of Chemistry and Biochemistry - Biochemistry, Berlin, Germany. [2] Berlin-Brandenburg School for Regenerative Therapies (BSRT), Berlin, Germany. [3] Leibniz-Forschungsinstitut für Molekulare Pharmakologie, Berlin, Germany. [4] Department of Biochemistry, University of Cambridge, Cambridge, UK. [5] These authors contributed equally: Jerome Jatzlau, Wiktor Burdzinski. ✉email: petra.knaus@fu-berlin.de

The transforming growth factor β (TGFβ) superfamily compromises more than 30 ligands, including bone morphogenetic proteins (BMPs) and activins, which regulate a broad range of developmental and homeostatic functions in a variety of cell types and organs, including the osteochondral and the vascular system[1,2]. TGFβ and BMP signal transduction is initiated by oligomerization of two type I and two type II serine/threonine kinase receptors upon binding one dimeric ligand. Trans-phosphorylation of the GS-box of a type I receptor by a type II receptor renders the type I receptor active, thereby allowing the recruitment and activation of SMAD proteins which act as transcriptional regulators[2].

TGFβ superfamily ligands exhibit preferential binding to either type I or type II receptors. BMPs were described to possess high affinity towards type I receptors, which induce SMAD1/5/8 signaling, e.g. BMP9 for ALK1, BMP6 for ALK2 and BMP2 for ALK3[3–5]. In contrast SMAD2/3 activating TGFβ superfamily members exhibit high affinity towards type II receptors, e.g. TGFβ1 for TGFBR2 or Activin A for ACVR2B[5–7]. However, some receptors are promiscuous in ligand binding (e.g. ACVR2B) and receptor expression levels vary in specific cell types. When high-affinity receptors are absent this often results in cell type-specific ligand competition scenarios between low-affinity receptors[8].

All TGFβ superfamily members are dimers that resemble two left hands with palms together, forming a butterfly-like structure. The dimeric ligand offers two 'wrist', 'knuckle', or 'fingertip' epitopes, which allows the binding of two type I and two type II receptors[9,10]. Indeed, structural variation in these ligands yields different binding modes of type I and type II receptors for the BMP, TGFβ, and Activin classes[8,11]. While both receptor types bind independently BMPs and Activins, TGFβ receptors bind cooperatively to TGFβs. Hereby TGFBR2 binds uniquely to the fingertips of TGFβ1–3, thus creating a shared epitope with the ligands for the low-affinity type I receptor ALK5 (i.e. cooperative binding)[10]. In contrast, ACVR2A, ACVR2B, and BMPR2 bind via hydrophobic interactions to the knuckle epitope[12–15]. High-affinity binding of BMP type I receptors to the wrist epitope can be attributed to a structurally unique α1 helix, which projects phenylalanine into a hydrophobic pocket of the ligands (e.g. ALK3) or forms polar bonds with both ligand monomers (ALK1), in a so-called lock-and-key mechanism[12,16,17]. Finally, Activins possess significant variability in the dimeric structure and are thought to become restrained upon type II receptor binding, adopting a conformation that together with the fingertips selects for type I receptor recruitment (i.e. conformational selection)[6,11,13,18].

Whereas co-crystallization of the ligands with their respective receptor extracellular domains (ECD) but also extensive surface plasmon resonance (SPR) experiments provided the structural framework and in vitro ligand–receptor affinities, a spatio-temporal resolution of the complexes with full-length membrane-bound receptors could so far only be achieved via microscopic approaches. Using immunofluorescence co-patching of epitope-tagged BMP receptors allowed the identification of preformed receptor complexes (PFCs) and BMP-induced signaling complexes (BISCs)[19]. The dynamic association of BMP receptors into homo- and heterodimers and the resulting lateral diffusion was shown by fluorescence recovery after photobleaching (FRAP) and Patch/FRAP experiments[19,20]. The dynamics of the short-lived BMP-induced heteromeric receptor complex formation were investigated by using two-color single particle tracking (SPT) utilizing quantum dots (QDs)[21]. However, most of the previous approaches relied on antibody recognition of epitope-tagged receptors, resulting in unknown fluorophore stoichiometries, requiring long incubation times and restricting the analysis to fixed cells. To overcome this, tagging receptors with self-labeling enzymes would allow for live-time analysis using straightforward staining protocols. Self-labeling protein tags like the SNAP- (derived from mammalian $O^6$-alkylguanine DNA alkyl transferase) or the Halo-tag (derived from bacterial haloalkane dehalogenase) specifically react with benzyl guanine (BG) or chloroalkane (CA), respectively, transferring one fluorophore of choice by means of covalent linkage, enabling fast and stoichiometric antibody-free labeling of a protein of interest (POI)[22,23]. Simultaneous application of both self-labeling protein tags for analysis of two POIs previously demonstrated a two-color visualization approach for intracellular targets[24]. The addition of sulfonate groups to SNAP- and Halo-tag ligands renders them cell-impermeable and thereby allows discrimination of receptor populations by exclusively staining cell surface exposed receptors[25,26].

In order to visualize all TGFβ/BMP receptors including their complexes in dual color we generated a comprehensive N-terminally Halo- and SNAP-tagged receptor library. Additionally, designing and applying fluorescently labeled growth factor ligands allowed us to not only monitor receptor-dependent ligand surface binding but also to establish a Ligand Surface Binding Assay (LSBA) for screening ligand binding under native conditions. As LSBA was suitable to test TGFβ superfamily ligand–receptor binding in living cells, this allowed us to provide additional proof for ligand–receptor interface specificities. To extend this, we designed and generated variants of type I and type II receptors with altered ligand specificities, as verified by microscopy. Further, we showed the general applicability of our system by for example comparing human receptor orthologs and distant teleost paralogs, providing a pre-testing platform before establishing animal models, thereby following the 3R principles. This work provides a new adaptable visualization platform for color multiplexing, suitable for super-resolution microscopy, and allowing to investigate ligand specificities to their respective receptor in living cells.

## Results

**Cell surface staining of Halo- and SNAP-tagged BMP-, Activin- and TGFβ-receptors.** BMPs, TGFβs, and Activins signal through binding towards different combinations of BMP/TGFβ type I and type II receptors in a hetero-tetrameric receptor–ligand complex (Fig. 1a). For example, the type I and the type II receptor ECD bind to the wrist and knuckle epitope of BMP family ligands, respectively (Fig. 1b). We here aimed to visualize preferential ligand–receptor binding on living cells using state of the art imaging tools. Limited visualization of BMP and TGFβ receptors by immunofluorescence can be overcome by utilization of self-labeling Halo- and/or SNAP-tag enzymes. BMP/TGFβ receptors, including type I receptors (hALK1-ALK6, rALK7, and hALK2-R206H) and type II receptors (hACVR2A, hACVR2B, hBMPR2, and hTGFBR2) were, therefore, N-terminally fused with Halo- or SNAP-tags separated by a 5x glycine linker downstream the signal peptide sequence, to obtain the receptor library shown in Fig. 1c. All Halo- and SNAP-tagged type I and type II receptor constructs were transiently expressed in COS-7 cells and lysates were subjected to Western blot analysis with specific antibodies directed against Halo- (Supplementary Fig. 1a) or SNAP-tags (Supplementary Fig. 1b). While untagged type I receptors have molecular weights between ~56 and 60 kDa, the Halo- or SNAP-tagged type I receptors were detected at a molecular weight corresponding to the addition of a Halo-tag (~33 kDa) or a SNAP-tag (~20 kDa) (Supplementary Fig. 1a, b). Similar weight shifts were observed for Halo- and SNAP-tagged type II receptors BMPR2, ACVR2A, ACVR2B, and TGFBR2

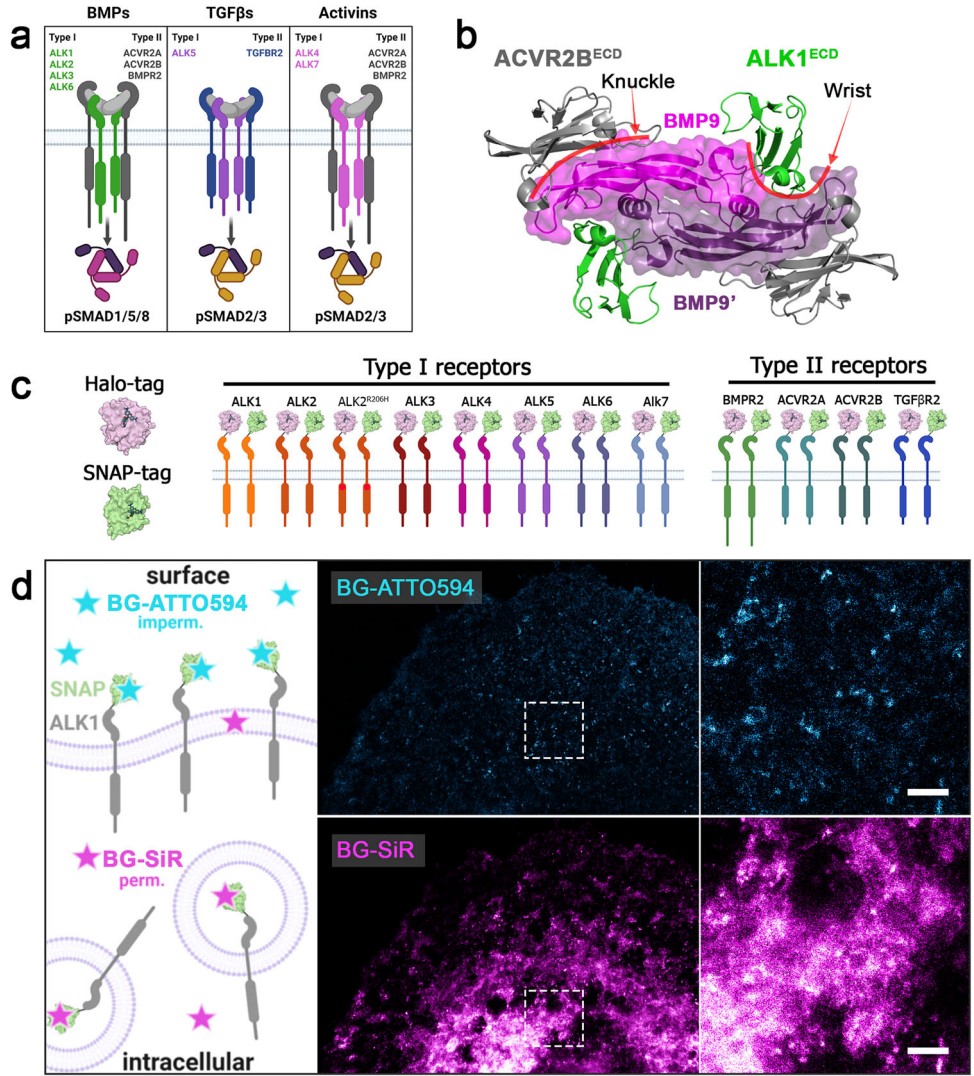

**Fig. 1 Halo- and SNAP-tagged BMP and TGFβ receptors can be visualized covalently and fast with organic fluorophore-labeled substrates allowing the discrimination between surface and intracellular receptor populations. a** Overview of preferential receptor-ligand paring. **b** Top view of the extracellular domains of ALK1 and ACVR2B in a heterotetrameric receptor complex bound to BMP9 (crystal structure PDB: 4FAO). Type I receptor ALK1 binds at the wrist interface, type II receptor ACVR2B binds at the knuckle interface of BMP9, respectively. **c** Schematic of N-terminally Halo- and SNAP-tagged BMP and TGFβ receptor library consisting of type I receptors (ALK1-ALK7 and FOP-mutant ALK2-R206H) and type II receptors (BMPR2, ACVR2A, ACVR2B, and TGFBR2). **d** Discrimination between surface and cytosolic receptor populations. COS-7 cells transiently expressing ALK1-SNAP were 24 h post-transfection incubated with BG-ATTO594 (impermeable; cyan) and chased by BG-SiR (permeable; magenta) allowing for staining of the surface receptor population and the cytosolic receptor population, respectively. BG benzyl guanine, SiR silicon rhodamine.

(Supplementary Fig 1a, b). Further, all Halo- and SNAP-tagged receptors were positively detected using impermeable CA-Alexa488 or BG-Alexa488 staining, respectively. Using cell-impermeable dyes provides the advantage to visualize exclusively the surface receptor pool, while cell-permeable dyes or fluorescent protein tags (e.g. GFPs) would stain both the membrane and intracellular fraction. As an example, we stained COS-7 cells transiently expressing ALK1-SNAP with first cell-impermeable SNAP-tag substrate BG-ATTO594, chased by cell-permeable BG-SiR, highlighting the advantage of surface-only staining (Fig. 1d). ATTO594 uniquely stained receptors located at the plasma membrane while SiR visualizes the crowded intracellular fraction. With the additional advantage that stimulated emission depletion (STED) compatible dyes were used, super-resolution imaging allowed a lateral resolution of ~40 nm pixel (Supplementary Fig. 1c). Together, this describes a comprehensive library of BMP and TGFβ receptors and showcases that CA-

and BG-coupled dyes that allow the exclusive study of the cell surface population by STED nanoscopy.

**High-affinity binding of Activin A by BMP type II receptors.** Specificity of Activin A binding to BMP/TGFβ receptors was previously mainly studied in cell-free SPR measurements utilizing immobilized receptor ECDs and soluble Activin A[13,18]. Binding and chemical crosslinking of [125]I-labeled Activin A in transfected COS-7 cells[27,28] allowed for the detection of ligand-bound receptors after cell lysis by radiography. Here, we aimed to visualize receptor-dependent Activin A binding on living transfected COS-7 cells in a fluorescence-based approach. Fluorescently labeled Activin A was recently used to show ACVR2B clustering[29]. Accordingly, we have labeled dimeric Activin A with Cy5 and established an analysis pipeline for assessment of Activin A-Cy5 binding on COS-7 cells dependent on the expression of

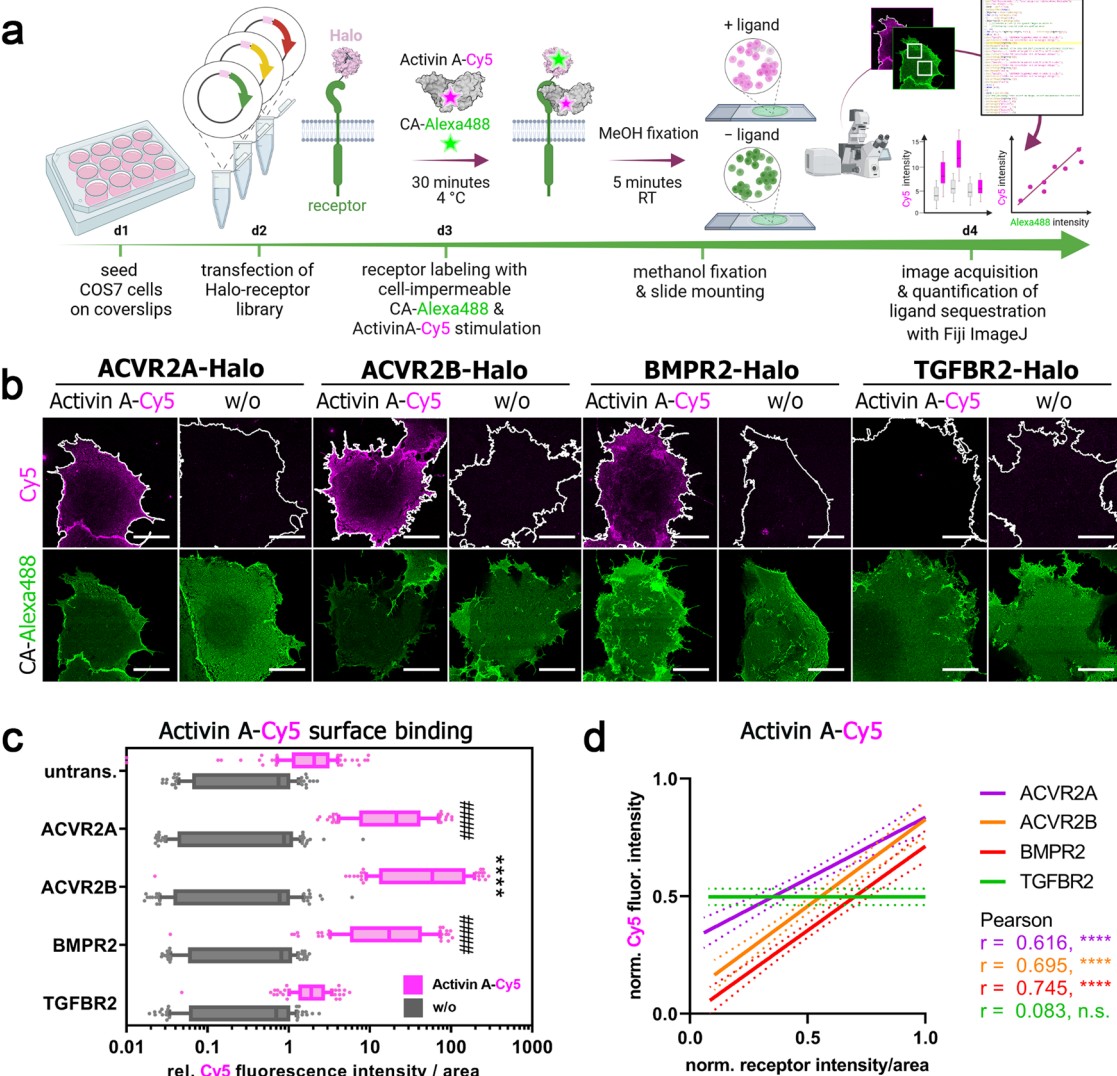

**Fig. 2 Activin A-Cy5 binding by BMP type II receptors. a** Schematic illustration for visualization of fluorescent growth factor binding on COS-7 cells expressing Halo-tagged BMP or TGFβ receptor constructs. **b–d** COS-7 cells were seeded on coverslips and transfected with indicated Halo-tagged receptor constructs. 24 h post-transfection, cells were incubated with non-permeable fluorescent Halo-tag substrate CA-Alexa488 (green) and Activin-Cy5 (magenta) for 30 min at 4 °C, fixated with methanol for 5 min at room temperature and mounted on glass slides. Cells were imaged at a confocal microscope and 10 cells per condition and replicate were analyzed with a semi-automated Fiji ImageJ macro pipeline for assessment of fluorescent growth factor binding (Activin A-Cy5) and fluorescence intensity of receptors (CA-Alexa488). Four ROIs of 100 $\mu m^2$ were quantified in each cell. Data shown are derived from three independent experiments. **b** Representative confocal microscopy images of COS-7 cells transiently expressing ACVR2A-, ACVR2B-, BMPR2- or TGFBR2-Halo incubated with CA-Alexa488 and simultaneously stimulated with Activin A-Cy5 or PBS as control. Scale bar ≙ 20 μm. **c** Activin A-Cy5 surface binding is represented as relative fluorescence intensity per area. Data is shown as FI ± SD. Significance was calculated using two-way ANOVA and Tukey's post-hoc test. #### $p < 0.0001$ ≙ significant relative to Activin A-Cy5 stimulated control cells, **** $p < 0.0001$ ≙ significant relative to all Activin A-Cy5 stimulated conditions including ACVR2A and BMPR2. **d** Linear regression and**d** correlation analysis of ligand:receptor binding based on Cy5-fluorescence intensity and normalized receptor fluorescence (CA-Alexa488) per area. CA: chloroalkane.

various Halo-tagged receptors (Fig. 2a). Receptors from the Halo-tagged BMP/TGFβ receptor library (Fig. 1c) were transiently expressed in COS-7 cells and stained with cell-impermeable Halo-tag substrate CA-Alexa488 at 4 °C to impede internalization. Simultaneous incubation with Activin A-Cy5 allowed for visualization of ligand binding to the respective receptor. Acquired confocal images were semi-automatically analyzed with Fiji ImageJ to quantify Activin A-Cy5 and receptor signal intensities (Fig. 2a). As expected ACVR2B led to the highest (~84-fold) Activin A binding, followed by ACVR2A (~29-fold) and BMPR2 (~27-fold), whereas TGFBR2 and all type I receptors showed no significant increase in Activin A binding (Fig. 2b, c, Supplementary Fig. 2a, b). Activin A-Cy5 and receptor intensities

positively correlated for ACVR2A, ACVR2B, and BMPR2, while TGFBR2 expression did not correlate with ligand binding, highlighting that increased high-affinity receptor expression allows more ligand binding (Fig. 2d). Similar results were obtained using U2OS cells, highlighting that LSBA can be performed in multiple cell types with different endogenous receptor expression levels (Supplementary Fig. 3). Live cell imaging (LCI) of COS-7 cells expressing ACVR2B-Halo highlighted that Activin A-Cy5 binding occurs already within minutes after ligand addition (Supplementary Movie 1 and Supplementary Fig. 4). Finally, we tested if FACS analysis could be employed for measurement of ligand binding. Indeed, high-affinity binding of Activin A to ACVR2B or ACVR2A- expressing cells- (~13.7-fold or ~8.9-fold, respectively)

was observed, whereas BMPR2-expressing cells (~2.8-fold) showed less binding and TGFBR2-expressing cells no binding. This highlights the general applicability of LSBA across different detection platforms, from which microscopic analysis provides higher sensitivity (Supplementary Fig. 5). Overall, these results are in accordance with binding and cross-linking studies in COS cells utilizing $^{125}$I-Activin A[27], as well as SPR studies using immobilized receptors[5] but reflect binding on living cells and under native conditions.

**SiR-d12 labeled-BMP9 and -TGFβ1 allow for the identification of their respective high-affinity receptors**. To extend the studies to other members of the TGFβ ligand superfamily, we labeled dimeric BMP9 and TGFβ1 with *N*-hydroxysuccinimide (NHS) silicon rhodamine-d12 (SiR-d12[30]) esters (Supplementary Fig. 6) to obtain far-red fluorescent BMP9-SiR-d12 and TGFβ-SiR-d12 (Fig. 3a; center). The labeling protocol is described in the "Methods" section and a scheme of SiR-d12-labeled ligands binding to their respective high-affinity Halo-tagged receptors and staining parameters are depicted in Fig. 3a (left and right). We next tested TGFβ1-SiR-d12 and BMP9-SiR-d12 binding on COS-7 cells, transiently expressing receptors found in signaling complexes of the respective ligand, i.e. TGFβ1-TGFBR2-ALK5 and BMP9-ACVR2B-ALK1[12,31]. As expected, we observed strong TGFβ1-SiR-d12 binding to cells expressing TGFBR2 but not to cells expressing ALK5 (Fig. 3b). Similarly, the BMP9-high affinity receptor ALK1 strongly increased binding of BMP9-SiR-d12, while ACVR2B, which was shown by SPR to exhibit a similar high picomolar affinity (ALK1-Fc $K_D$ = 31.3 pM, ACVR2B-Fc $K_D$ = 33.0 pM)[12], failed in binding BMP9-SiR-d12 (Fig. 3b, c). Both TGFβ1-SiR-d12 and BMP9-SiR-d12 signal intensity positively correlated to TGFBR2 or ALK1 receptor intensity, respectively (Fig. 3d). Finally, we expanded our ligand surface binding assay to the whole receptor library but did not observe any other significant increases for either TGFβ1-SiR-d12 or BMP9-SiR-d12 binding by any other receptor (Fig. 3c, d, Supplementary Figs. 7, 8). Structure prediction of Halo-tagged receptors highlights no interference of growth factor:receptor interface for any of the tested ligands, due to long flexible Gly5 linkers (Supplementary Fig. 9a, b). Equally, we confirmed that Halo-tagged receptors remain signaling competent (Supplementary Fig. 9c–f).

**Receptor-mimics reveal critical residues for Activin A–type II receptor interfaces**. Binding of Activin A-Cy5 to cell surface expressed type II receptors confirmed a higher affinity of ACVR2B over ACVR2A and BMPR2, which showed comparable ligand binding (Fig. 2)[5]. Structural analysis of Activin A bound to ACVR2B previously highlighted that the core of the concave ACVR2B interface contains hydrophobic residues, which interact with hydrophobic residues in the Activin A knuckle epitope, surrounded by polar residues at the edge that form strong electrostatic interactions (Fig. 4a)[13]. Sequence comparison of ACVR2B with ACVR2A indicated only subtle differences in two amino acids which form hydrophobic bonds with Activin A (Tyr$_{A2B}$60/Phe$_{A2A}$61, Phe$_{A2B}$82/Ile$_{A2A}$83), whereas BMPR2 is characterized by a different elongated 2/3 loop, which harbors mostly polar and charged amino acids of the Activin A epitope in ACVR2B (Leu$_{A2B}$79-Asp$_{A2B}$80-Asp$_{A2B}$81-Phe$_{A2B}$82-Asn$_{A2B}$83) (Fig. 4a). Evolutionarily conserved residues and potentially crucial for ligand binding properties in this area include Trp78 (Fig. 4a). We next asked if substituting half of the BMPR2 2/3 loop (Ser$_{BR2}$84-X-Gln$_{BR2}$90) with the Activin A-binding ACVR2B amino acids (Leu$_{A2B}$79-X-Asn$_{A2B}$83) would increase ligand binding of BMPR2. The resulting BMPR2$^{Mimic-ACVR2B}$-Halo construct was generated (Supplementary Fig. 10a) and Activin A-Cy5 binding

was analyzed as described above (Fig. 2a, Supplementary Fig. 10b). Indeed, BMPR2$^{Mimic-ACVR2B}$ showed increased binding compared to BMPR2 and even to ACVR2B, highlighting the crucial role of the 2/3 loop in Activin A ligand binding (Fig. 4b–d). We next asked whether the altered 2/3 loop in BMPR2$^{Mimic-ACVR2B}$ forms similar interactions with Activin A residues as ACVR2B. We, therefore, took advantage of the ACVR2B/Activin A crystal structure (PDB: 1S4Y)[18] and the crystallized ECD of BMPR2 bound to Activin B (PDB: 7U5O)[15,32] and performed in silico docking of the BMPR2 structure to Activin A (Supplementary Fig. 11). Next, we generated a homology model of BMPR2$^{Mimic-ACVR2B}$ docked on to Activin A. Indeed, our predictions highlight that BMPR2$^{Mimic-ACVR2B}$ forms the same electrostatic interactions as ACVR2B with Activin A at its 2/3 loop (most importantly a salt bridge between Asp$_{A2B}$80 with Arg$_{ActA}$87) (Fig. 4e), while in BMPR2 these interactions are altered (e.g. Ile$_{BR2}$88 forms a less favorable interaction with Arg$_{ActA}$87). Collectively this data highlighted that our Halo-tagged receptor library is suitable for assessment of ECD mutation screening and that the substitution of the 2/3 loop of BMPR2 with the ACVR2B 2/3 loop yields an Activin A high-affinity BMPR2.

**Generation of a BMP9-binding ALK2 receptor-ALK2$^{Mimic-ALK1}$**. After successfully increasing the affinity of a type II receptor (Fig. 4), we aimed to equally enhance the binding of BMP9-SiR-d12 to type I receptors. For this, we selected ALK2-Halo to substitute key residues from the high-affinity receptor ALK1. In line with our data (Fig. 3b, c), ALK1 is the high-affinity receptor for BMP9[3,12] and presents with a ~1800 times higher affinity (ALK1-Fc $K_D$ = 45.5 pM) when compared to ALK2 (ALK2-Fc $K_D$ = 83 nM)[4]. However, ALK2 was shown to transmit BMP9 signals in the absence of ALK1 in non-endothelial cells[4,33–35]. The ALK1–BMP9 interface is subdivided into three sites (I–III). Site I and III are overlapping with other type I receptor–BMP interfaces while site II is unique to ALK1 for BMP9 and BMP10 binding[4,12]. Alongside, the helix α1 and loop F3 of ALK1 contain charged residues (Fig. 5a), which through polar interactions convey binding to both BMP9 monomers within the wrist epitope[4,12]. These polar residues (HERR motif— His73, Glu75, Arg78, and Arg80) bridge to amino acids that are unique to BMP9 and BMP10, thereby defining ligand specificity[4]. Sequence comparison with ALK2 highlighted that the HERR motif surrounding the helix α1 is absent in ALK2, alongside other critical residues that define ALK1/BMP9 specificity (Fig. 5a). We, therefore, generated three ALK2 variants, in which we sequentially substituted amino acids by the ALK1/BMP9 epitope (V1–3 ALK2$^{Mimic-ALK1}$) and tested BMP9-SiR-d12 surface binding in transfected COS-7 cells (Fig. 5b–d, Supplementary Fig. 12a, b). Interestingly, the substitution of the HERR motif only was not sufficient to increase BMP9-binding to ALK2 (V1 ALK2$^{Mimic-ALK1}$), however additional substitution of hydrophobic residues surrounding the HERR motif in the helix α1 rendered ALK2 BMP9-sensitive (V2 ALK2$^{Mimic-ALK1}$). Additional substitution of amino acids at the site I interface (VVFREE-motif, V3 ALK2$^{Mimic-ALK1}$) did not further increase ligand binding. In line with increased BMP9 binding, V2 and V3 ALK2$^{Mimic-ALK1}$ expression elevated BMP9-induced BRE-Luc activity similar to ALK1-Halo (Fig. 5e). To elucidate possible interactions of the ALK2 variants, we designed a homology model of each variant and docked it to BMP9 based on the ALK1/BMP9/ACVR2B complex (PDB:4FAO) and AlphaFold2 predictions of ALK2 (Supplementary Figs. 13, 14, Fig. 5f)[12,36,37]. Structure comparison underlined the incapability of the ALK2 helix α1 to form strong bonds at the wrist epitope (Fig. 5f). However, helix α1 of V2 but not of V1

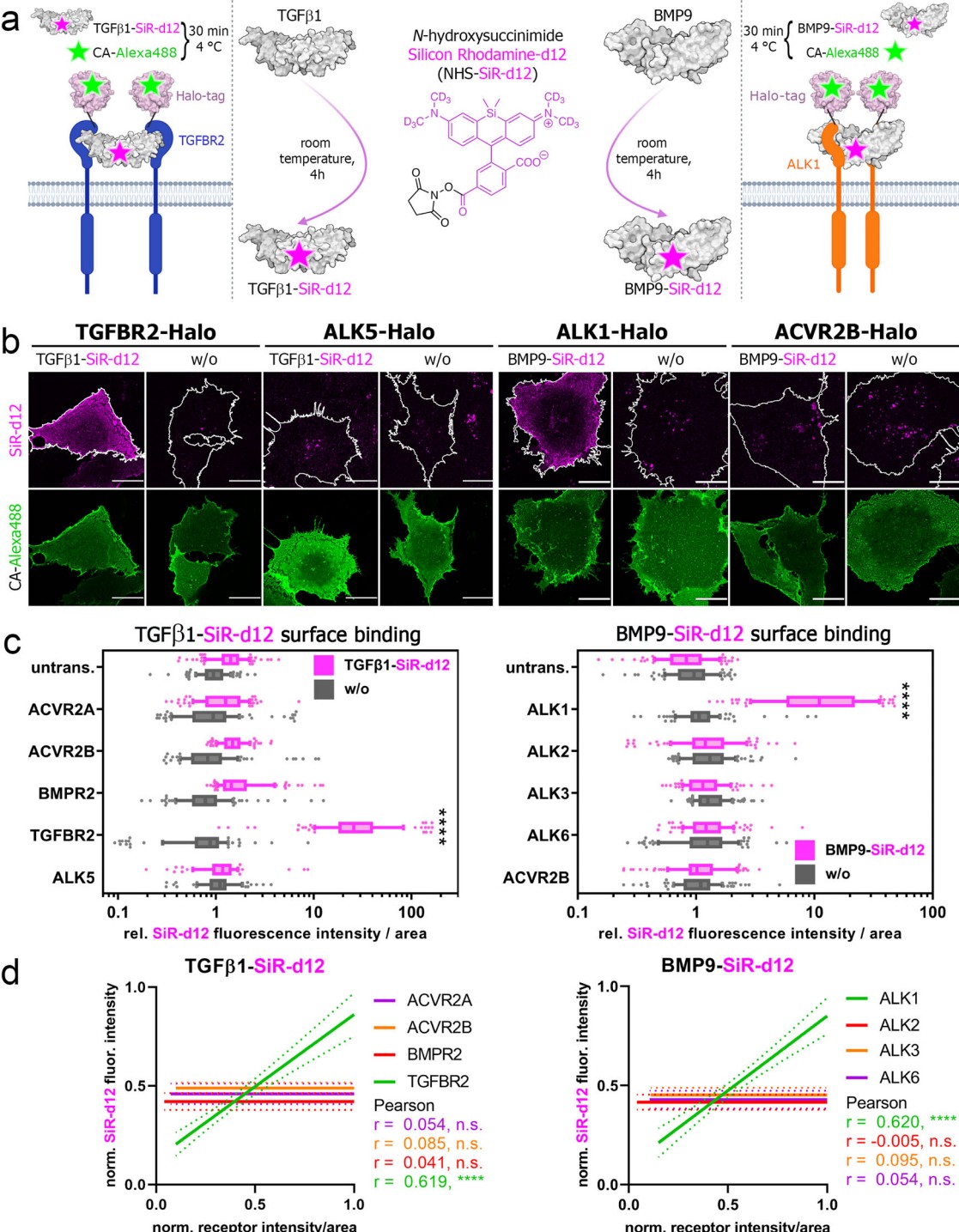

**Fig. 3 Silicon Rhodamine-d12 labeling of TGFβ1 and BMP9 visualizes binding by TGFBR2 and ALK1, respectively. a–d** Transiently transfected COS-7 cells expressing TGFBR2-, ALK5-, ALK1- or ACVR2B-Halo were 24 h post-transfection simultaneously incubated with Halo-tag substrate CA-Alexa488 (green) and TGFβ1-SiR-d12 or BMP9-SiR-d12 (magenta). The data shown are derived from three independent experiments. **a** Ligand labeling strategy of TGFβ1 and BMP9 utilizing $N$-hydroxysuccinimide deuterated silicon rhodamine ester (NHS-SiR-d12) (middle). Illustration of TGFβ1-SiR-d12 and BMP9-SiR-d12 binding to their respective high-affinity receptor TGFBR2 (left) or ALK1 (right). **b** Representative confocal microscopy images of TGFβ1-SiR-d12 (left) and BMP9-SiR-d12 stimulated COS-7 cells expressing respective high- (TGFBR2, ALK1) or low-affinity receptors (ALK5, ACVR2B). Scale bar ≙ 20 μm. **c** TGFβ1-SiR-d12 (left) and BMP9-SiR-d12 (right) surface binding represented as relative fluorescence intensity per area. Data is shown as FI ± SD. Significance was calculated using two-way ANOVA and Tukey's post-hoc test. ****$p < 0.0001$ ≡ significant relatives to all stimulated conditions. **d** Linear regression and correlation analysis of ligand:receptor binding based on SiR-d12-fluorescence intensity and normalized receptor fluorescence (CA-Alexa488) per area ($n = 3$). CA chloroalkane.

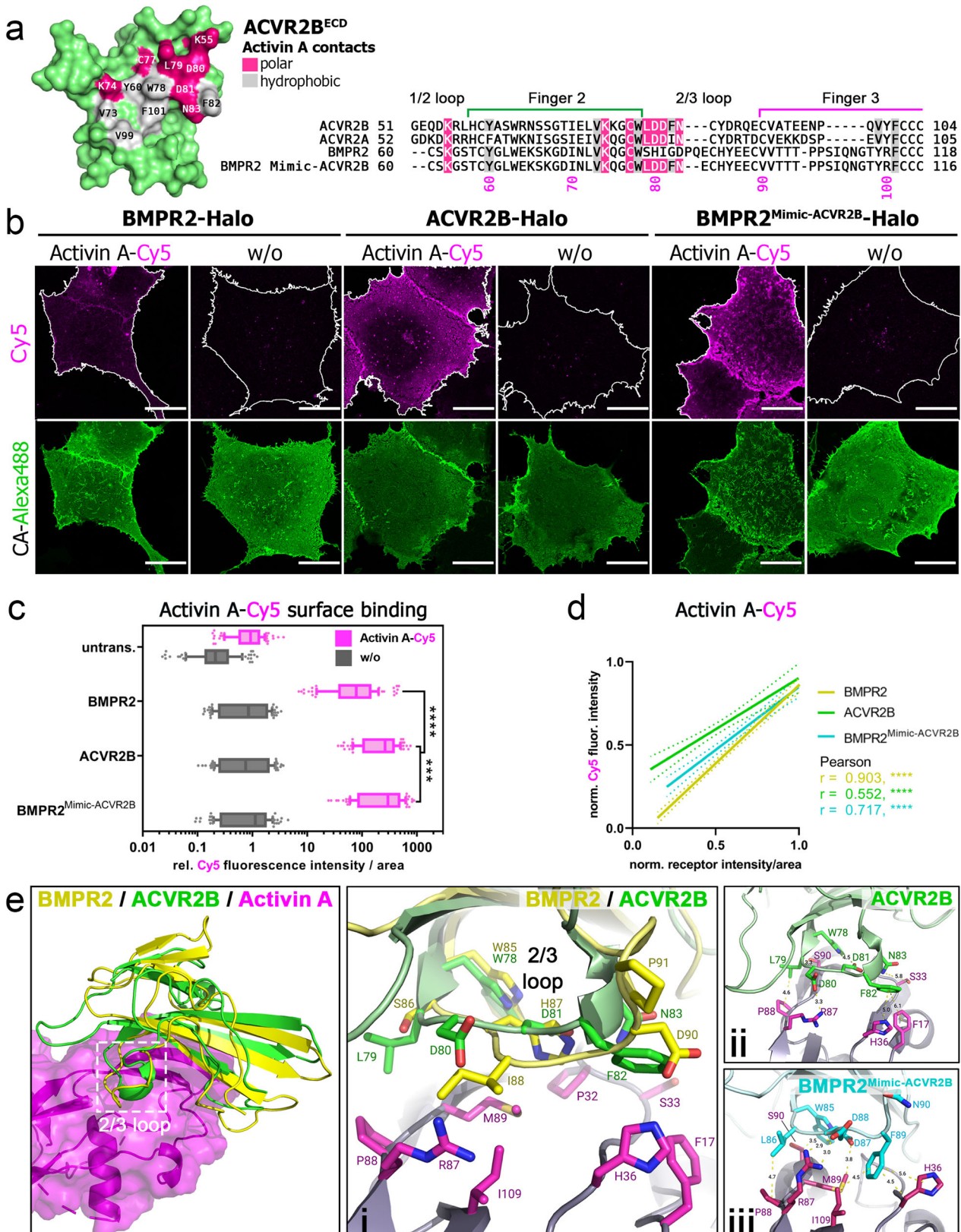

ALK2[Mimic-ALK1] can form interactions with BMP9, similarly as in ALK1 (Fig. 5f, Supplementary Fig. 14b). Substitution of the HERR motif in V1 ALK2[Mimic-ALK1] is likely insufficient due to the rigidity and/or altered conformation of surrounding residues (e.g., Thr[V1]79 *versus* Gly[V2/ALK1]79), prohibiting the conformations of the mutated residues needed for interacting with BMP9 (Supplementary Fig. 14b). Although the binding interface of V3 ALK2[Mimic-ALK1] and BMP9 was expected to resemble the ALK1 binding interface more closely, the F2 loop differs in its conformation when comparing ALK1 and V3 ALK2[Mimic-ALK2], possibly due to the ALK2-derived overall fold and surrounding residues (Supplementary Fig. 14c). This likely explains the

**Fig. 4 BMPR2 gains Activin A affinity through substitution of ACVR2B 2/3 loop. a** Molecular surface of ACVR2B[ECD] showing the Activin A interface (PDB: 1S4Y). Residues forming polar or hydrophobic contacts are colored in pink and gray, respectively (left). Sequence alignment of BMP type II receptors with Activin A contact residues highlighted. BMPR2[Mimic-ACVR2B] sequence includes ACVR2B substitutions in the 2/3 loop. **b–d** Transiently transfected COS-7 cells expressing BMPR2-, ACVR2B- or BMPR2[Mimic-ACVR2B]-Halo receptors were 24 h post-transfection simultaneously incubated with Halo-tag substrate CA-Alexa488 (green) and Activin A-Cy5 (magenta). The data shown are derived from three independent experiments. **b** Representative confocal microscopy images of COS-7 cells transiently expressing BMPR2-, ACVR2B- or BMPR2[Mimic-ACVR2B]-Halo incubated with CA-Alexa488 and simultaneously stimulated with Activin A-Cy5 or PBS as control. Scale bar ≙ 20 μm. **c** Activin A-Cy5 surface binding represented as relative fluorescence intensity per area. Data is shown as FI ± SD. Significance was calculated using two-way ANOVA and Tukey's post-hoc test. \*\*\*$p < 0.001$ \*\*\*, $p < 0.0001$ ≡ significance as indicated ($n = 3$ independent experiments). **d** Linear regression an**d** correlation analysis of ligand:receptor binding based on Cy5-fluorescence intensity and normalized receptor fluorescence (CA-Alexa488) per area ($n = 3$). **e** Overview (left) and cartoon/stick representation (i) of the binding interface of the ACVR2B/Activin A (PDB: 1S4Y) crystal structure compared to our in silico predicted BMPR2/Activin A complex (structures derived from PDB: 7U5O and 1S4Y, respectively). Distance measurements (using PyMOL) in Angstrom of the 2/3 loop residues of ACVR2B (ii) and BMPR2[Mimic-ACVR2B] variant (iii) to Activin A reveal possible interactions. Docking predictions were carried out using the Rosetta docking protocols.

observed low binding of V3 ALK2[Mimic-ALK1] to BMP9 (Fig. 5c). Together this indicates that exchanging critical residues of the helix α1 of ALK2 with ALK1 residues transfers BMP9 sensitivity.

**Determining BMP9-binding to ALK1 teleost orthologs.** Finally, labeled ligands are a useful tool to study receptor binding in in vivo systems. Nonetheless, ligand–receptor binding should and can be tested before going into animal experiments using LSBA. As an example, we tested BMP9 binding to different teleost (subcohort of Clupeocephala) ALK1 orthologs (Fig. 6). Both zebrafish (*Danio rerio*) and medaka (*Oryzias latipes*) are two well-characterized fish model systems and distant vertebrate relatives of humans (Fig. 6a). However, zebrafish and medaka ALK1 have a shortened F2 loop and unique pre-helix loops which differ from the human ALK1. Overall, in medaka 25% and in zebrafish only 20% of the human BMP9/ALK1 interface is conserved (Fig. 6b). Interestingly, binding of human BMP9 was also observed for both teleost ALK1s but at a reduced level compared to human ALK1 (Fig. 6c–e). Interestingly, transient expression of zebrafish or medaka ALK1 in HEK cells elevated BMP9-induced BRE-Luc activity to the same extent as via the human ALK1, suggesting the necessity of endogenous type II receptors for BMP9-signaling via the SMAD pathway (Fig. 6f, Supplementary Fig. 12c), which covers the different affinities to the single expressed respective type I receptors. In summary, LSBA is also applicable for pre-testing of animal models and evolutionary studies.

## Discussion

Interrogating cell surface receptor biology in living cells, including their distribution, stoichiometries, ligand-binding efficacies, and downstream activation has been pushed in the last years mainly by state-of-the-art microscopy techniques[38,39]. In all these instances, the choice of labeling the protein of interest is of immense importance[40]. While traditional fluorescent protein tagging is powerful, it comes with several drawbacks, such as delay in chromophore maturation, observation of intracellular signals, limited choice of different colors (especially in the far-red spectrum), and reduced performance in super-resolution nanoscopy due to bleaching. Small organic dyes overcome this issue, which are more robust and brighter and can be selectively targeted in a temporal manner to self-labeling enzymes, such as SNAP- and Halo-tags. Spatial resolution can furthermore be achieved by using membrane-impermeable substrates and dyes that restrict labeling to extracellularly exposed tags. Even more, the possibility to use orthogonal tags allows the use of different colors, rapidly adding to the portfolio of multiplexed experiments to visualize receptor pairs and their ligand interaction in live cells. Recent examples include the determination of SNAP-mGluR4 co-localization (with $Ca_v2.1$ and Bassoon) by two-color dSTORM

microscopy[39] and the compositional diversity of differently tagged-kainate receptor assembly by three-color single molecule TIRF microscopy[41]. Alternatively, visualization of cell surface ligand binding, endo- and exocytosis, as well as receptor clustering was achieved using ligands covalently linked to fluorophores[29,42–46]. For this, ligands were either directly labeled at deprotonated primary amines using NHS-activated fluorophores[42,43], or through the introduction of an N- or C-terminal cysteine residue allowing for site-directed maleimide coupling of the dyes[45,46].

Herein, we combined N-termin ally Halo- and SNAP-tagged BMP and TGFβ receptors with fluorescently labeled growth factors that allowed us to measure and to visualize ligand binding to cell surface receptors in living cells. Whereas ligand–receptor affinities are commonly assessed using in vitro methods such as surface plasmon resonance measurements, we attempted to measure ligand binding on living cells with the ability to resolve subcellular ligand binding events. Applying this *Ligand Surface Binding Assay* (LSBA) allowed for binding measurements under more physiological conditions with a focus on the plasma membrane. While in this study we focused on the binding properties of single receptors, future studies can include additional binding partners. Whereas crosslinking experiments using radio-labeled growth factors have been performed as one of the first methods to define high- and low-affinity BMP/TGFβ receptor affinities[7,47,48], the use of fluorescently labeled ligands increases the field of application to multiple microscopy techniques. By using labeled Activin A it was recently shown that ligand-mediated receptor clustering depends on the presence of ACVR2A/B[29]. Earlier studies using labeled BMP2 gave insights into ligand uptake and release[43–45].

To test whether N-terminally Halo-/SNAP-tagged receptors are still capable of ligand binding we performed the LSBA using different members of the TGFβ superfamily of ligands (Activin A, TGFβ1 and BMP9), which are known to possess high affinities towards different type I or type II receptors. Our findings from LSBA measurements using Activin A-Cy5 are in good agreement with previous binding and crosslinking studies using radiolabeled ligands, highlighting that only the BMP/Activin type II receptors are capable of Activin A binding and they do not require the presence of type I receptors[27,28,49]. SPR analysis confirmed that ACVR2B[ECD] exhibited the highest affinity ($K_D = 1.1$ nM) followed by ACVR2A[ECD] ($K_D = 5.7$ nM) and BMPR2[ECD] ($K_D = 59$ nM)[5]. Receptor dimerization will further enhance Activin A binding as indicated by SPR studies using ACVR2A-Fc ($K_D \sim 130$ pM) and BMPR2-Fc ($K_D \sim 14$ nM)[49], which was further confirmed by a comparison of Activin A binding to a single ECD ACVR2B-Fc ($K_D = 108$ pM) versus a homodimeric (ACVR2B)$_2$-Fc ($K_D = 12.5$ pM)[6]. Interestingly, BMPR2–Activin A complexes have been suspected to be more transient due to

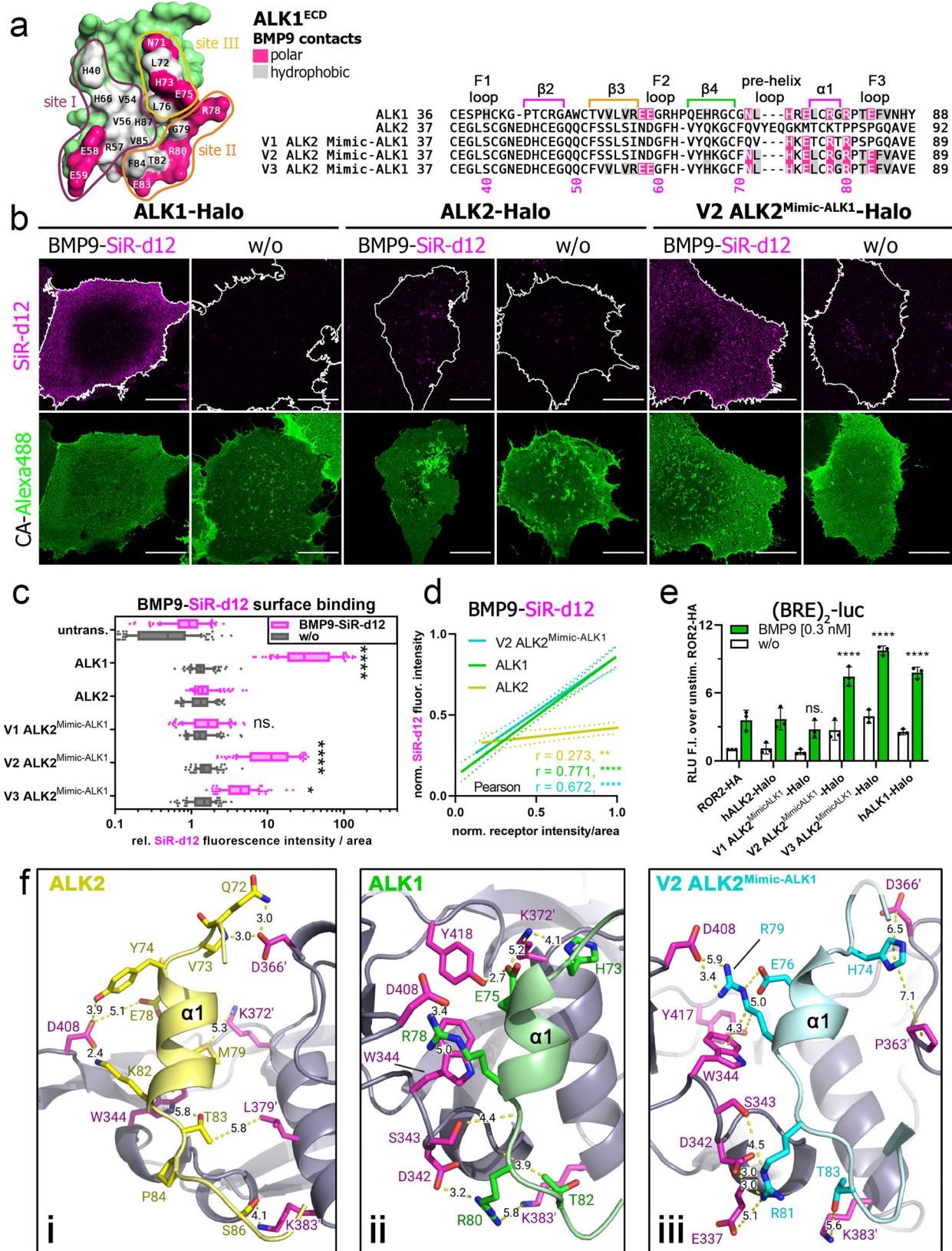

**a** ALK1^ECD BMP9 contacts: polar, hydrophobic. site I, site II, site III labeled. Sequence alignment of ALK1, ALK2, V1 ALK2 Mimic-ALK1, V2 ALK2 Mimic-ALK1, V3 ALK2 Mimic-ALK1 showing F1 loop, β2, β3, F2 loop, β4, pre-helix loop, α1, F3 loop.

**b** ALK1-Halo, ALK2-Halo, V2 ALK2^Mimic-ALK1-Halo with BMP9-SiR-d12 and w/o; SiR-d12 and CA-Alexa488 rows.

**c** BMP9-SiR-d12 surface binding. **d** BMP9-SiR-d12 Pearson: r = 0.273 **; r = 0.771 ****; r = 0.672 ****. **e** (BRE)₂-luc.

**f** ALK2 (i), ALK1 (ii), V2 ALK2^Mimic-ALK1 (iii).

their higher dissociation rate, compared to ACVR2A-Activin A[49]. These differences are confirmed by sequence comparison of the ACVR2B–Activin A interface with ACVR2A and BMPR2 ECD sequences. While ACVR2A shares mostly the same 2/3 loop with ACVR2B, which contains polar and charged amino acids that are critical for Activin A binding, BMPR2 has an alternative 2/3 loop[13]. We could show that substitution of these critical residues elevated Activin A binding to BMPR2 by ~3 fold. In the same light, substitution (H87A) of a sterically unfavorable Histidine located in the 2/3 loop led to a 1.5-fold increase of ^125I-Activin A

**Fig. 5 ALK2 gains BMP9 affinity through substitution of ALK1 helix α1. a** Molecular surface of ALK1$^{ECD}$ showing the BMP9 interface (PDB: 4FAO). Residues forming polar or hydrophobic contacts are colored in pink and gray, respectively (left). Interaction sites I–III are indicated with colored lines. Sequence alignment of ALK1, ALK2, and variants V1–3 of ALK2$^{Mimic-ALK1}$ with BMP9 contact residues highlighted. ALK2$^{Mimic-ALK1}$ sequences include sequential ALK1 substitutions at interaction sites II and III (V1–2) and site I (V3). **b–d** Transiently transfected COS-7 cells expressing ALK1-, ALK2-, V1–V3 ALK2$^{Mimic-ALK1}$-Halo receptors were 24 h post-transfection simultaneously incubated with Halo-tag substrate CA-Alexa488 (green) and BMP9-SiR-d12 (magenta). The data shown are derived from three independent experiments. **b** Representative confocal microscopy images of COS-7 cells transiently expressing ALK1-, ALK2- or V2 ALK2$^{Mimic-ALK1}$-Halo incubated with CA-Alexa488 and simultaneously stimulated with BMP9-SiR-d12 or PBS as control. Scale bar ≙ 20 μm. **c** BMP9-SiR-d12 surface binding represented as relative fluorescence intensity per area. Data is shown as FI ± SD. Significance was calculated using two-way ANOVA and Tukey's post-hoc test. $****p < 0.0001 \equiv$ significant relative to BMP9-SiR-d12 stimulated control cells. **d** Linear regression and correlation analysis of ligand:receptor binding based on SiR-d12-fluorescence intensity and normalized receptor fluorescence (CA-Alexa488) per area. **e** After one day of transfection with the SMAD1/5/8-sensitive BRE$_2$-luciferase reporter and RLTK-Luc together with ALK1-Halo, ALK2-Halo or ALK2$^{Mimic-ALK1}$-Halo (V1-3), HEK293t cells were starved for 5 h and stimulated with BMP9 (0.3 nM) overnight. Relative Luminescence Units (RLU) are expressed as mean fold induction ± SD ($n = 3$ independent experiments). Statistical significance relative to unstimulated ROR2-HA-transfected control cells was calculated using two-way ANOVA and Tukey's post-hoc test. **f** Cartoon and stick representation of the "HERR"-motif residues (α1 helix and F3 Loop) of ALK2 (based on an AlphaFold model docked to BMP9 (PDB: 4FAO) using Rosetta docking protocols) (i), ALK1 co-crystallized to BMP9 (PDB: 4FAO) (ii), and V2 ALK2$^{Mimic-ALK1}$ (iii) (derived from ALK2, mutated, and docked to BMP9 (PDB: 4FAO) using the Rosetta Commons Modeling Suite). ALK2 cannot form interactions originating from the helix α1 and surrounding loops, while residues of ALK1 and V2 ALK2$^{Mimic-ALK1}$ can form alike interactions to BMP9 (interaction sites II and III) as shown by stick representation and distance measurements (using PyMOL).

surface binding in CHO cells[49]. Together this highlights that the LSBA is suitable for mutant screening and assessment of high-affinity ligand binders.

We next performed fluorescent labeling of BMP9 and TGFβ1, using a recently reported, state-of-the-art silicon rhodamine-d12 (SiR-d12) as its NHS ester, which exhibits high brightness and increased lifetime, and as such is suitable for confocal, life-time and super-resolution microscopy[30]. Both BMP9-SiR-d12 and TGFβ1-SiR-d12 bound their known high-affinity receptor, i.e. ALK1 and TGFBR2, respectively[3,7]. Interestingly, whereas BMP9 is supposed to signal via ALK2 in non-endothelial cells that lack ALK1 expression[4,33–35], we did not observe BMP9 binding by any other type I receptor except ALK1. This is in line with the described difference of ALK1 being ~1800 times more sensitive to BMP9 than ALK2 as determined by SPR[4]. BMP9 is secreted into the bloodstream from stellate cells in the liver[50]. In the vasculature BMP9 circulates at concentrations (2–12 ng/mL) that are ~100 times higher than the EC$_{50}$ of BMP9 to ALK1 (EC$_{50}$ = 50 pg/mL, 2 pM)[3,51], highlighting that endothelial cells, which express predominantly ALK1 as type I receptor, are exposed to saturating BMP9 conditions. BMP9 signaling via ALK2 was reported to play a role in a variety of non-endothelial cells[52]. Mechanistically, this was supported by radioactive crosslinking studies which observed BMP9–ALK2 binding in myoblasts and breast cancer cells[53]. Further, the authors highlighted that co-expression of BMP type II receptors in COS-7 cells enhances BMP9 binding to ALK2. While local BMP9 expression has been described in the developing murine CNS[54], the predominant source of BMP9 in the human body is circulating BMP9 originating from the liver[51]. Therefore, ALK2-dependent BMP9 signaling in non-endothelial cells could play a prominent role in the context of leaky or injured vessels accompanied by blood spillage into the tissue. In line with multiple reports of BMP9 influencing tumor progression[52], tumors are characterized by immature leaky blood vessels[55]. Similarly, BMP9 could fulfill its previously described pro-osteogenic function[56] in the context of bone fracture healing.

Most of the ALK1–BMP9 interface is unique[12], which is why we sequentially substituted key residues of ALK2 with corresponding residues of ALK1. Particularly substituting the amino acids of the pre-helix loop and the helix α1 was sufficient to increase BMP9 binding to V2 ALK2$^{Mimic-ALK1}$. Interestingly, we did not detect ACVR2B dependent BMP9 binding, while SPR studies using ACVR2B-Fc and ALK1-Fc indicated a similar affinity of BMP9 for both receptors[12]. A potential explanation could be that a stable dimer of ACVR2B, as present in the Fc-fusion

protein, is required for the high-affinity binding of BMP9, which might not be the case for the full-length receptors at the plasma membrane. While a comparative study for ACVR2B binding to BMP9 has to our knowledge not been performed, monomeric ALK1-ECD (K$_D$ ~ 71.6 pM) possesses a lower affinity to BMP9 compared to dimerized ALK1-Fc (K$_D$ ~ 48.1 pM)[4]. It is unlikely that BMP9 binding to ACVR2B was negatively influenced by the N-terminal Halo-tag, as the same receptor was able to bind Activin A normally[4,12,13]. However, an independent SPR study highlighted that the affinity of Activin A (K$_D$ ~ 9.2 pM) towards ACVR2B-Fc is eleven times higher than the affinity of BMP9 (K$_D$ ~ 116.1 pM)[57]. Furthermore, administration of ACVR2B-Fc as a GDF8/Myostatin inhibitor (ACE-031) in a clinical trial to treat Duchenne muscular dystrophy had to be stopped due to the occurrence of telangiectasias[58], typical of deficient BMP9-signaling in the vasculature. Together, this highlights that BMP9 efficiently binds to monomeric ALK1 at low concentrations, as well as to hetero- (e.g. ALK2/ACVR2B) or homodimeric receptor complexes (ACVR2B/ACVR2B).

Whereas we have shown that the LSBA assay is a valuable method to study surface ligand sequestration of high-affinity ligand:receptor pairs, a current limitation of the assay is the capability to measure ligand binding mediated by heterodimeric receptor complexes. Future studies, therefore, have to address how to quantify the effect of type I and type II co-expression, which would require equal expression levels of both. Also, it will be interesting to expand the receptor library towards known and novel BMP- and TGFβ-co-receptors and subsequently test, if ligand binding to the receptors is dependent on the presence of respective co-receptors. Since all tested high-affinity ligand:receptor pairs recapitulated previously reported binding behaviors, a negative influence of N-terminal receptor-tagging on ligand-binding could be excluded. While N-terminal Halo-tags in combination with non-permeable dyes allow the exclusive study of receptors at the cell surface, C-terminal tagging in combination with permeable Halo-dyes would allow the visualization of also internal receptor pools. Further, the future development of smaller self-labeling enzymatic tags would be a great advancement in the field. Comparing ligand binding by both microscopic and FACS analysis highlighted that both methods have individual advantages. While the microscopic LSBA is capable to detect endogenous ligand binding and subtle differences in ligand-binding at subcellular resolution and in a time-dependent manner, FACS analysis allows the unbiased quantification of a high number of cells. A disadvantage of FACS-based analysis is the

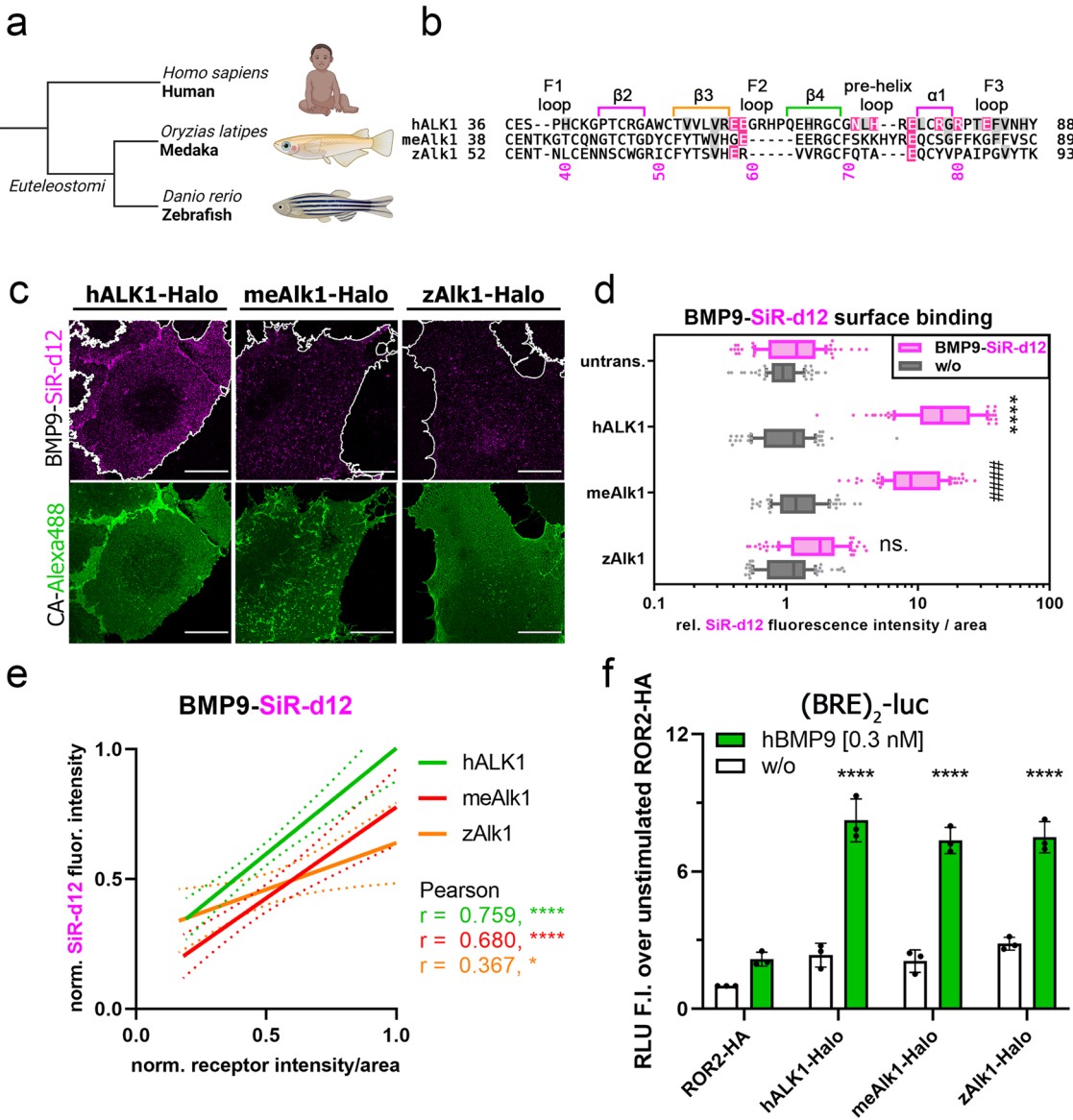

**Fig. 6 BMP9 binding to ALK1 is conserved in vertebrates. a** Medaka (*Oryzias latipes*) and zebrafish (*Danio rerio*) are distant vertebrate relatives of humans. **b** Sequence alignment of human, medaka, and zebrafish ALK1 with human BMP9/ALK1 contact residues are highlighted. **c–e** Transiently transfected COS-7 cells expressing human, medaka, or zebrafish ALK1-Halo receptors were simultaneously incubated with Halo-tag substrate CA-Alexa488 (green) and BMP9-SiR-d12 (magenta) 24 h post-transfection. The data shown are representative of $n = 3$ experiments. **c** Representative confocal microscopy images of COS-7 cells transiently expressing different Halo-tagged ALK1 paralogs incubated with CA-Alexa488 and simultaneously stimulated with BMP9-SiR-d12 or PBS as control. Scale bar ≙ 20 µm. **d** BMP9-SiR-d12 surface binding represented as relative fluorescence intensity per area. Data is shown as FI ± SD. Significance was calculated using two-way ANOVA and Tukey's post-hoc test. ****$p < 0.0001 ≡$ significant relative to BMP9-SiR-d12 stimulated control cells. **e** Linear regression and correlation analysis of ligand:receptor binding based on SiR-d12-fluorescence intensity and normalized receptor fluorescence (CA-Alexa488) per area. **f** After one day of transfection with the SMAD1/5/8-sensitive BRE$_2$-luciferase reporter and RLTK-luc together with human, medaka, or zebrafish ALK1-Halo receptors, HEK293t cells were starved for 5 h and stimulated with BMP9 (0.3 nM) overnight. Relative luminescence units (RLU) are expressed as mean fold induction ± SD ($n = 3$ independent experiments). Statistical significance relative to unstimulated ROR2-HA-transfected control cells was calculated using two-way ANOVA and Tukey's post-hoc test.

requirement of an enzymatic step for cell detachment, which could deteriorate receptor:ligand binding. The LSBA on the other hand can be performed on fixed and living cells without cell detachment. Since our current studies were limited to three ligands, each representative for one TGFβ ligand sub-family, it will be intriguing to expand the analysis to other members of the TGFβ superfamily, and potentially include heterodimeric ligands such as BMP9/BMP10. Therefore site-directed labeling strategies in combination with dual-color labeling might be required to efficiently assess hetero-dimeric ligand binding.

In summary, we established an imaging platform that allows us to revise and test ligand–receptor affinities in a cellular context. Due to the nature of the self-labeling Halo- and SNAP-tag, BMP and TGFβ receptors can now be visualized at a stoichiometric ratio using modern dyes, which meet the requirements for the respective experiments, e.g impermeability for surface-only stainings, increased photo-stability including STEDability[26,30]. Future studies can now address (1) differences in ligand–receptor binding of monomeric receptors vs. receptor complexes, (2) binding behavior of other TGFβ-superfamily ligands using the

incorporation of unnatural amino acids at the C-terminus allowing site-directed coupling and positioning of fluorophores, and (3) visualization of endogenous receptor populations in living cells and in tissues using fluorescently labeled high-affinity ligands as a replacement for insufficient antibodies.

## Methods

**Restriction cloning**. First, hBMPR2-LF-SNAP/Halo and mUnc5b-SNAP/Halo were generated by substitution of meGFP in hBMPR2-LF-meGFP and mUnc5b-meGFP (*doctoral thesis J. Jatzlau*) with SNAP or Halo using Sec-61-Halo and pSNAPf as a template[24]. For the generation of the N-terminally Halo- and SNAP-tagged BMP/TGFβ receptor library, hALK1-hALK6, rALK7,hALK2-R206H as well as mAlk1 and zAlk1 were subcloned into pcDNA3.1 mUnc5b-SNAP/Halo and hACVR2A, hACVR2B, hTGFBR2 were subcloned into pcDNA3.1 hBMPR2-LF-SNAP/Halo in between EcoRI and NotI thereby replacing the respective receptor ORF. Receptor encoding sequences with respective restriction site overhangs have been amplified using the primers listed in Supplementary Table 1. All cloning PCRs were carried out on a Peltier Thermal Cycler PTC-200 and the primers used are listed in (Appendix Table 1). The elongation time was adapted according to the product size (Phusion Pol. = 1 kb/min). PCR products were resolved by agarose gel electrophoresis and purified using NucleoSpin Gel and PCR Clean-up kit, according to the manufacturer's guidelines. For restriction cloning, PCR products and 3 μg of the destination vector were digested overnight with 1 μL (~20 Units) of each restriction enzyme. Successful restriction digest was validated on an agarose gel, from where cleaved products were purified. Subsequently, 50 ng destination vector and 300 ng insert were ligated for 10 min at room temperature with T4 ligase (~40 Units) in 10 μL H₂O. After 5 min heat inactivation at 65 °C, the total volume of ligation mix was added to 50 μL chemically competent DH5α or TOP10 E. coli and incubated for 30 min on ice. After a 90 s heat shock in a 42 °C water bath, the bacteria were put back on the ice for 2 min. Next, the bacteria were shaken for 1 h at 37 °C and 180 rpm in 1 mL LB medium w/o antibiotics. Finally, bacteria were plated on LB-agar plates with antibiotics and grown overnight at 37 °C. On the next day, 3 mL LB medium with the respective antibiotics was inoculated with bacterial colonies and grown at 37 °C and 180 rpm overnight, followed by plasmid DNA purification and Sanger sequencing.

**Cell culture**. COS-7 and U2-OS cells were obtained from the German Collection of Microorganisms and Cell Cultures (DSMZ) and cultured in Dulbecco's modified Eagle's medium (DMEM) supplemented with 10% FCS, 2 mM ʟ-glutamine, and penicillin (100 units/mL)/streptomycin (100 μg/mL) (DMEM full medium) in a humidified atmosphere at 37 °C and 5% CO₂ (v/v). COS-7 and U2-OS cells were maintained in T175 flasks and cells were split 1:3 or 1:5, depending on need, and were kept sub-confluent. For passaging, cells were washed once with PBS before being removed from the flasks surface with trypsin/EDTA (0.05/0.02% in PBS).

**Transient transfection with expression plasmids**. For microscopy studies of COS-7 and U2-OS cells, cells were transfected with Lipofectamine 2000 according to the manufacturer's instructions. 100,000 cells/12-well were seeded in 1 mL DMEM full medium. On the following day, cells in each well were transfected with a total amount of 500 ng DNA. The subsequent experimental procedure took place 24 h post-transfection unless indicated otherwise.

**SDS–PAGE and Western-blotting**. For sodium dodecyl sulfate–polyacrylamide gel electrophoresis (SDS–PAGE), treated cells were lysed in 150 μL Laemmli buffer and frozen at −20 °C. The lysate was pulled through an 18-gauge syringe and boiled for 10 min at 95 °C. 10% polyacrylamide gels were cast in advance and stored at 4 °C until usage. Separated by their molecular weight, proteins were transferred onto methanol-activated PVDF membranes by Western-blot. Membranes were blocked for 1 h in 0.1% TBS-T containing 3% w/v BSA, washed three times in 0.1% TBS-T and incubated with indicated primary antibodies overnight at 4 °C. Primary antibodies: anti-GAPDH (Cell Signaling; #2118; monoclonal rabbit antibody), anti-Halo (ProMega; #G9211; monoclonal mouse antibody), and anti-SNAP (Invitrogen; #CAB4255; polyclonal rabbit antibody) were applied at a 1:1000 dilution in 3% w/v bovine serum albumin (BSA)/fraction V in TBST. For HRP-based detection, goat-α-mouse or goat-α-rabbit IgG HRP conjugates (±0.8 mg/mL, Dianova; #111-035-144, #115-035-068) were used at a dilution of 1:10,000. Chemiluminescent reactions were processed using WesternBright Quantum HRP substrate (advansta) and documented on a FUSION FX7 digital imaging system.

**Halo- and SNAP-tagged receptor staining and ligand visualization**. For visualization of COS-7 and U2-OS cells transiently expressing Halo- and SNAP-receptor fusion proteins, 200,000 cells were transfected on glass coverslips (confocal: 18 mm ø or STED: precision #1.5H 10 mm ø) with desired constructs. For live cell imaging, 100,000 COS-7 cells were seeded on glass-bottom culture dishes (35 mm ø). 24 h post-transfection, cells were washed once with PBS and incubated, depending on the experimental setup, with a fluorescent SNAP- and/or a fluorescent Halo-ligand in the absence or presence of fluorescent growth factors for

30 min. The following commercially available cell-impermeable fluorescent ligands were utilized: HaloTag Fluorescent Ligand Alexa Fluor 488 (Promega, #G1002) in the following abbreviated as CA-Alexa488; SNAP-Surface 594 (NEB, #S9134S) in the following abbreviated as BG-ATTO594 and SNAP-Cell 647-SiR (NEB, # S9102S), in the following abbreviated as BG-SiR. Surface receptor staining was carried out at 4 °C, whereas incubation at 37 °C took place during studying growth factor binding on living cells. Cells were washed once with PBS, before fixation with 100% methanol for 5 min. Cells were washed with PBS once more and mounted with Fluoromount G (confocal microscopy).

**Preparation of fluorescently labeled recombinant Activin A**. Mature dimeric Activin A was prepared by refolding from inclusion bodies as described before for Atto-647 and CF640R-labeled activin A[29] with the exception that mature activin A was untagged and the label used here was NHS-activated Cy5 (Lumiprobe, cat. no 13020). NHS-Cy5 was dissolved in DMSO to a concentration of 3 mM and lyophilized activin A dissolved in 10 mM HCl to 763 μM (10 mg/mL). The protein:dye ratio in labeling was 1:2 and final concentrations were 76 μM Activin A and 153 μM NHS-Cy5 in 50 mM HEPES pH 7.4, 30% acetonitrile in the total volume of 125 μL. The labeling reaction was incubated for 3 h at 23 °C and excess unreacted dye was quenched by adding Tris and ethanolamine to a final concentration of 8 mM each. The labeling sample was acidified by diluting the sample to 2 mL with 0.1% trifluoroacetic acid and loaded into ACE5 C8 300 4.6 × 250 mm reversed-phase chromatography column equilibrated with 10% acetonitrile, 0.1% tri-fluoroacetic acid. The protein was eluted with a linear gradient to 90% acetonitrile, 0.1% trifluoroacetic acid in 40 mL with a flow rate of 1 mL/min. Protein was detected at 280 and 646 nm (Cy5 excitation maximum). Fractions with Cy5 absorbance were analyzed for labeling efficiency spectrophotometrically using Cy5 molar extinction coefficient at 646 nm of 250,000 L mol⁻¹ cm⁻¹, activin A protomer extinction coefficient at 280 nm of 18,050 L mol⁻¹ cm⁻¹ and abs₂₈₀ₙₘ/abs₆₄₆ₙₘ correction factor for Cy5 of 0.04 (LumiProbe product data sheet) to account for dye absorbance at 280 nm. Fractions with at most 1:1 ratio of dye:activin protomer were pooled, divided into 5 μg aliquots, dried in black tubes using a centrifugal vacuum concentrator, and stored at −80 °C.

**Cell stimulation with growth factors**. rhBMP9/GDF2 and rhTGFβ1 (PeproTech, Hamburg, Germany) were reconstituted and stored according to the manufacturer's instructions. For microscopical binding studies, COS-7 cells were stimulated with 2 nM Activin A-Cy5. For microscopical binding studies (COS-7) of SiR-d12- or labeled growth factors, cells were stimulated with 0.3 nM TGFβ1-SiR-d12 and 0.3 nM BMP9-SiR-d12 if not indicated otherwise.

**Fluorescent growth factor labeling and Ligand Surface Binding Assay (LSBA)**. For fluorescent labeling of dimeric BMP9 and TGFβ1, N-hydroxysuccinimidyl deuterated silicon rhodamine (NHS-SiR-d12) (Supplementary Note 1) was taken up in DMSO to a final concentration of 1 mM. Lyophilized rhBMP9 (Peprotech) and rhTGFβ1 (Peprotech) were reconstituted in 0.2 M sodium bicarbonate (reaction buffer), pH = 8.3, to obtain a final concentration of 2 μM. NHS-SiR-d12 was then added to the reconstituted ligands at a 5-fold molar excess. Final concentrations being at 2 μM (Protein) and 10 μM (NHS-Dye). The mixture was allowed to incubate at room temperature for 4 h. Meanwhile, a 0.5 mL Amicon Ultra 10 kDa molecular weight cutoff column (Merck Millipore, UFC501024) was calibrated with reaction buffer and centrifuged at 14,000×g at 4 °C without drying the filter. The reaction mixture was then gently applied onto the pre-calibrated column and centrifuged for 4 min, while never allowing the filter to dry. Subsequently, the filter was gently washed with reaction buffer *ad* 500 μL and centrifuged as described before. This washing procedure was repeated twice. Afterward, 50 μL sterile Millipore H₂O was added to the column, before the column was inverted and put in a fresh elution tube and centrifuged as described above. The column was rinsed with 30 μL sterile Millipore H₂O and eluted as in the previous step. Protein concentration was determined with a NanoDrop 2000 spectrophotometer (Thermo Fischer). For visualization of SiR-d12-ligand binding on COS-7 cells, cells were transfected as described above with corresponding high- and low-affinity Halo receptors. The day after, cells were washed with PBS and, additionally to fluorescent Halo-ligand CA-Alexa488, simultaneously incubated with fluorescent growth factors at previously indicated concentrations for 30 min at 4 °C. Subsequently, cells were washed once with ice-cold PBS before fixation with 100% methanol for 5 min. Cells were washed again with PBS and mounted with Fluoromount G (confocal microscopy) or ProLong Gold Antifade Mounting reagent (STED microscopy).

**Fluorescence-activated cell sorting (FACS)**. For FACS analysis of ligand binding towards COS-7 cells transiently expressing Halo-receptor fusion proteins, 600,000 cells were transfected in six wells with desired constructs. 24 h post-transfection, cells were placed on ice and washed twice with ice-cold DPBS followed by incubation with fluorescent Halo-ligand CA-Alexa488 (0.5 μM) in the absence or presence of Activin A-Cy5 (2 nM) for 30 min at 4 °C. Subsequently, cells were washed once with ice-cold DPBS before incubation with Accutase solution at RT for 15 min. After detachment, cells were resuspended in DPBS and centrifuged at (1000 n/min) for 5 min. Finally, cells were resuspended in PBS (100 mM EDTA)

and measured in BD LSRFortessa™ Cell Analyzer. FACS data were analyzed using the BD FACSDiva 8.0.1 software. Cells were gated first for single cells using SSC-A/FSC-A, followed by the gating of 1000 transfected cells with a high signal in the FITC channel. Activin A-Cy5 surface binding was quantified as the mean Cy5 intensity of AF488$^+$ cells.

**Confocal and STED microscopy**. Confocal and stimulated emission depletion (STED) data of fixed or living COS-7 cells were acquired with the Expert Line STED Microscope from Abberior. Confocal images of COS-7 cells expressing Halo-tagged receptors stained with CA-Alexa488 and fluorescent ligands (Cy5- and SiR-d12-labeled) were acquired using 485 nm (20% laser power) and 640 nm excitation (20% laser power). STED images were acquired using 561 nm excitation for ATTO 594 (20% laser power) and 640 nm excitation for SiR (2% laser power). A 775 nm STED laser at 10% laser power was used to deplete both dyes. Live cell confocal imaging of COS-7 cells transiently expressing ACVR2B-Halo was carried out at an Abberior Expert Line STED microscope at a physiological temperature of 37 °C using a stage-top incubator (okolab). Cells were stained with 0.5 μM Halo-tag substrate CA-AlexaFluor488 (Promega) for 30 min at 37 °C, washed extensively, and kept on ice in HEPES buffer pH 7.4 until imaging. Focussing on transfected and stained cells, Activin A-Cy5 was added to the cells at 2 nM and dual-color images were acquired for 2 min every 2 s. Confocal and STED Images were acquired with the following settings in the Abberior Imspector Acquisition & Analysis Software v16.3: objective lens: ×100 NA1.4 (oil) [UPLS], pinhole: 1.0 AU, range: 75 μm × 75 μm, confocal pixel size: 60 nm, STED pixel size: 20 nm, pixel dwell time: 5/10 μs.

**Image analysis and semi-automated quantification with Fiji ImageJ**. Confocal raw data were post-processed and adjusted for color and contrast (linear adjustments maintained for confocal datasets represented within one figure) using Fiji (ImageJ) software and Adobe Photoshop (Adobe Systems). Surface binding quantification of Activin A-Cy5, BMP9-SiR-d12, and TGFβ1-SiR-d12 on COS-7 cells transiently expressing SNAP- & Halo-Receptor constructs was performed with Fiji. Per cell, four regions of interest (ROI) (100 μm²) were chosen and the raw integrated intensity (RawIntDen) of each ROI was measured both in receptor and ligand channels. Per condition, 10–30 cells were quantified in three independent experiments. Receptor–ligand binding was calculated relative to intensity values of untransfected COS-7 cells, representing endogenous ligand–receptor binding. Linear regression and correlation analysis of RawIntDen values of receptors and ligands was performed and plotted in GraphPad Prism 8.0 (GraphPad Software Inc.). Maximum RawIntDen values of receptors were normalized to 1. All used scripts are available under https://github.com/Habacef/Supplementary-scripts-for-LSBA (https://doi.org/10.5281/zenodo.7389579)[59].

**Dual-Luciferase Reporter gene Assay**. HEK293T cells were transfected with indicated Halo-tagged and HA-tagged Type I and/ or Type II receptors together with the SMAD1/5/8 sensitive (BRE)$_2$ or SMAD2/3 sensitive (CAGA)$_{12}$ luciferase reporter. A constitutively expressing construct encoding renilla luciferase (RL-TK; Promega) was co-transfected as an internal control. The next day, cells were starved in serum-free medium for 5 h before stimulation with BMP9 (0.2 nm) for 24 h. Cell lysis was performed using passive lysis buffer (Promega) and measurement of luciferase activity was carried out according to the manufacturer's instructions using a TECAN initiate f200 Luminometer (TECAN).

**Statistics and reproducibility**. All statistical tests were performed using GraphPad Prism version 9.3 software and are listed in the figure legends. The normal distribution of data sets was tested with the Shapiro–Wilk normality test. In cases of failure to reject the null hypothesis, the ANOVA and Tukey's post hoc test was used to check for statistical significance under the normality assumption. For all experiments, statistical significance was assigned, with an alpha level of $p < 0.05$. All Box-plots are defined as: center line, median; box limits, 10–90 quartiles; whiskers, 1.5× interquartile range; points, outliers. Each experiment was performed at least three independent times on different days ($n = 3$).

**Homology modeling**. The receptor variant BMPR2$^{Mimic-ACVR2B}$ was modeled starting from a BMPR2 x-ray structure (PDB: 7U5O)[15]. The receptor variants V1 ALK2$^{Mimic-ALK1}$, V2 ALK2$^{Mimic-ALK1}$ and V3 ALK2$^{Mimic-ALK1}$ were modeled starting from the AlphaFold2 prediction of ALK2[36,37], as the topology of the ECD of ALK2 is not yet experimentally solved. The starting structures were prepared by adding missing loops and residues using the GreedyOptMutationMover[60,61] (if applicable) followed by coordinate-constrained relaxation protocol to obtain optimized H-bonds based on the Rosetta Energy Function 2015(ref2015)_cst score function within RosettaScripts[62]. The GreedyOptMutationMover was also used to generate 500 replicas of each of the variants via insertion of mutations (and if applicable a short, up to 3 AA long, loop modeling). Within the same RosettaScript, a full FastRelax[63,64] with the filters "BuriedUnsatHBonds" and "PackStat" was performed for each generated variant. The best variants were discriminated by plotting the RMSD of all heavy backbone atoms vs. the Rosetta-Score (using scoring function ref2015) of the final 500 variants. The RMSD was calculated against the input structures (either BMPR2 or ALK2) using the RMSDMetric mover[65] in RosettaScripts. The 10 variants with low REU-values (score) and low RMSD were visually inspected, and the most reliable structure based on convincing intramolecular interactions and comparison to known BMP-receptor-ECDs was used for receptor–ligand-docking. The described plots together with cartoon representations of all variants are shown in Figs. S11a, S13a. All used RosettaScripts can be found in an open-source repository under https://github.com/Habacef/Supplementary-scripts-for-LSBA (https://doi.org/10.5281/zenodo.7389579)[59].

**Receptor ligand docking**. All homology models of receptor variants, but also the X-ray structure of BMPR2 and the AlphaFold prediction of ALK2 were docked to Activin A and BMP9, respectively. BMPR2 and BMPR2$^{Mimic-ACVR2B}$ were superimposed to Activin A-bound ACVR2B (X-ray structure PDB: 1S4Y). ALK2, V1 ALK2$^{Mimic-ALK1}$, V2 ALK2$^{Mimic-ALK1}$ and V3 ALK2$^{Mimic-ALK1}$ were superimposed to BMP9-bound ALK1 (X-ray-structure PDB: 4FAO)[12]. As the receptor binding interfaces (Fig. S11a) are known, local Monte-Carlo-based protein-protein docking could be performed in 500 replicas. Here, first, the "Docking" mover was used with low-resolution (backbone plus centroid) flags before a high-resolution (full atom) docking (using ref2015 scoring function). Other parameters as distance perturbation and angle perturbation were left on default. The Interface RMSD (IRMSD) was plotted against the score of the interface residues (ISC). Here too, the best 10 hits showing the lowest IRMSD and lowest ISC were screened for the final docked homology model. The described plots together with cartoon representations of all variants are shown in Figs. S11b, S13b. All used scripts are available under https://github.com/Habacef/Supplementary-scripts-for-LSBA (https://doi.org/10.5281/zenodo.7389579)[59].

**Graphical schemes and figures**. Figures were created with BioRender.com and *PyMOL*, Available at http://www.pymol.org/pymol.

(Schrödinger, L. & DeLano, W., 2020.)

For additional information regarding materials and methods, see the Appendix Supplementary Methods.

**Reporting summary**. Further information on research design is available in the Nature Portfolio Reporting Summary linked to this article.

## Data availability

The data supporting the findings of this study are available from the corresponding author upon request. Uncropped and unedited blot/gel images are shown in (Supplementary Fig. 15). Numerical source data of shown graphs are shown in Supplementary Data 1.

## Code availability

All used scripts described in this study are available under https://github.com/Habacef/Supplementary-scripts-for-LSBA (https://doi.org/10.5281/zenodo.7389579)[59].

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

## Acknowledgements

We thank Ramona Birke for her technical assistance. J.J. was supported by the Deutsche Forschungsgemeinschaft DFG (BSRT, SFB958) and the Einstein Center ECRT. W.B. was supported by an Einstein Kickbox grant for young scientists from the Einstein Center for

Regenerative Therapies. P.K. acknowledges the support from Deutsche Forschungsgemeinschaft DFG (FOR2165; SFB1444), the Einstein Center ECRT, Morbus Osler Society, and BMBF (PrevOP-Overload). We would like to acknowledge the assistance of the Core Facility BioSupraMol FU Berlin supported by the DFG. F.B. was supported by the Deutsche Forschungsgemeinschaft DFG (SFB958).

## Author contributions

J.J. and P.K. designed the study; J.J., J.B., and P.K. designed the labeling strategies and evaluated the data; W.B., J.J., and M.T. performed the experiments; L.O. performed receptor homology modeling & docking calculations; K.Ro. and J.B. produced NHS-SiR-d12; K.Ra. and M.H. produced the Cy5-labeled Activin A; F.B. provided expertise in STED microscopy; J.J., W.B., and P.K. wrote the manuscript; all authors commented on the manuscript.

## Funding

## Competing interests

The authors declare no competing interests.
