## [Peer Review File · Communications Biology]

Reviewers' comments:

Reviewer #1 (Remarks to the Author):

The manuscript by Jatzlau et al describes the design and production of a library of TGF β type 1 and type 2 serine/threonine kinase receptors with N-terminal Halo- and SNAP-tags. These chimeric receptors are able to self-label with the addition of specific chemicals linked to fluorophores, providing a platform for measuring co-localization of the ligand with the receptor. The authors proceed to utilize the library to measure binary interactions of labeled TGF β ligands with the receptors and perform specificity experiments where modifications to the receptors are analyzed for altered ligand specificity. Overall, this paper functions largely as a methodology paper, describing a specific experimental tool and providing some examples of how this system might be used as a form of validation. The tool itself is novel, filling a gap in the field's ability to explore ligand-receptor interactions in a living cell. This is in part, due to the lack of antibodies that specifically bind to each receptor that do not interfere with ligand binding. Accordingly, it might result in important advances in the understanding of growth factor-receptor interactions across the TGF β signaling pathway (as hinted in the final paragraph of the discussion). With this in mind, the manuscript falls short of advancing the field in a major direction. Can the platform be used to measure both type I and type II receptors at the same time or is it only useful for high affinity interactions? Collectively, this leaves the reader wondering what are the capabilities and limitations of the system. Thus, significant modification would be needed to improve the manuscript.

Specific comments

- The results are highly dependent on consistent expression of receptors. Truncated gel analysis makes it difficult to interpret if there are any aggregation of receptors, or engineered receptors. Since the receptors have been modified with certain tags that can be recognized via FACS, I would request the authors analyze and normalize all results with a more quantitative measurement such as FACS toward either tag.
- The manuscript was prepared by multiple primary authors. While this is frequently the case, here noticeable differences between quality of writing, experimental thoroughness, and rigor of reporting between sections are apparent. This is particularly noticeable when comparing section 2 and section 3 of the results section, and the corresponding methodology. The preparation and labeling of BMP9 and TGF β 1 are thoroughly described, while this is less true for Activin A. There was some experimental validation that Halo-tagged receptors could still function in a separate experimental system (BRE luciferase reporter assay) for ALK1 and ALK2. However, this was not performed for ACVR2B. More consistency between sections would improve the manuscript.
- Overall, the validation experiments of this system were lacking. This is of particular importance when trying to quantify what is essentially a qualitative assay system, like microscopy. While I appreciate the BRE luciferase confirmation that the tagged receptors can induce signaling, it would have been nice to compare these results to WT receptor (no tag). An alternative would have been to compare ligand binding affinities between WT and tagged receptors using orthogonal/biophysical approaches. While the relative mean fluorescent readings did overall agree with the reported literature (ActA binds more strongly to ACVR2B in both this system and in literature reports of SPR), a more direct comparison would have been appreciated. In particular, this could have answered some of the discrepancies the authors observed in areas where their observations did not match up with the literature, such as the interaction between BMP9 and ACVR2B (Results Section 3, Fig. S7).
- In section 6, human BMP9 binds to ALK1 from different teleost species that do not resemble human ALK1 across any of the residues that were deemed important to binding in section 5. Does this finding discount the findings in section 5? Does this imply that there are multiple possible binding sites/orientations? This point is mentioned in the text of the manuscript, but could the authors provide a possible explanation?.
- Figure 5 E is not all that convincing. The error bars for the different version of Alk2 that show

binding are very large. The background is increasing considerably. This data does not seem strong enough to support that different version of the swaps of Alk1 into Alk2 are responding to BMP9 treatment. More rigorous analysis would be better, including titrations.

- The Western-blot validation of construct expression levels is inconsistent. ALK7 never really seems to express well (Fig. S1). Of more concern, ALK2-Halo does not seem to express at similar levels to all other receptors in Fig. S9, calling into question whether the conclusions of section 5 are even applicable. Maybe there was little binding of BMP9 to the cells transfected with ALK2-Halo because the transfection didn't work. Maybe the reason that there was less binding of BMP9 to V3 ALK2Mimic-ALK1-Halo is that less of it expressed than of the other ALK2 mutants. Again, these results should somehow be normalized, perhaps thorough FACS.
- For consistency, Western-blot analysis of receptor transfection into HEK293T cells, in addition to COS-7, would have helped inform the results of the BRE luciferase experiments.
- What is the endogenous expression of these receptors in COS-7 cells? What effect, if any, might this have on the overexpression of these tagged receptors?
- The fact that BMP9 binding to WT ALK2 is not detected (as described in Line 305-322) is attributed to relatively lower affinity between BMP9 and ALK2 as compared to ALK1. This implies that this system is only useful for detecting high affinity ligand:receptor interactions, limits the utility of this system (i.e. what are the limitations of the system?). It would have been interesting to determine whether increasing concentrations of labeled BMP9 might have been detectable.

Minor comments

- There is a lack of consistency in verb tense used throughout this document. For example: "We here aim..." (Line 104) vs. "Here, we aimed..." (Line 133). This is present throughout the Introduction and Results sections and should be corrected.
- Reword line 144-147 as correlation is not causation. (i.e. not leads to more ligand binding)
- Change "...dyes allow to exclusively study..." to "dyes that allow the exclusive study of..." (Line 126).
- Line 197 links to Fig. S6C. There is no Fig. S6C, only A and B.
- In Fig. 6D, the graph is missing the bars for untreated cells transfected with the different ALK1 paralogues.
- In Fig. S4, it would also be nice to have this data for ActA-Cy5.

Reviewer #2 (Remarks to the Author):

In this manuscript Jatzlau and co-workers describe an elegant assay making use of differentially tagged receptors of the TGFb/BMP superfamily in combination with fluorescently labelled ligands for optical quantification of receptor-dependent ligand binding. This is a well designed and well conducted study providing a valuable assay to evolve our understanding of the TGFb superfamily and its downstream effectors. There are some issues however that need further clarification.

1. In a family of signaling molecules where expression levels and ratios of receptors will influence function and outcome, how does overexpression influence the outcome?
2. Most of the studies have been performed in COS-7 cells expressing certain members of the TGFb family on their cell surface. Would the outcome be different when different cell types were used?
3. As TGFbeta type 3 receptors modulate affinity as well, would it be possible to include them in the assay?
4. TGFb ligands bind a dimers to the receptor complex. Is the labelled ligand monomeric or dimeric?
5. Some of the effects are related to heterodimeric proteins composed of e.g. BMP9/BMP10. Can the authors speculated if this can be included in the assay our would make interpretation and translation more difficult.

Reviewer #3 (Remarks to the Author):

The current understanding of ligand-receptor binding and assembly in the TGF- β family has mainly been achieved from structural studies and SPR-based binding studies with purified ligands and receptor ECDs. The new approach proposed by Jatzlau and co-workers, which uses HALO- and SNAP-N-terminally tagged type I and type II receptors, extends this by allowing one to study assembly as it occurs on the surface of live cells. Thus, with this approach, it should be possible to study all ligand-receptor interactions occurring simultaneously with a ligand, rather than just one at a time. This is important as this can extend understanding to include effects arising from membrane localization, receptor competition, co-receptor binding, among other possible effects.

The results presented focused on studies of some singly-tagged receptors interacting with a given target ligand. The tagged receptors are all clearly synthesized and reach the cell surface and imaging of these, with fluorescently labeled ligands, nicely recapitulates previous binding measurements for ActA, TGF- β 1, and BMP-9. The measurements also appear to be sufficiently sensitive to detect relatively modest gains of affinity, as demonstrated through structure-based engineering of BMPRII and Alk2. The technical quality of the work is outstanding, and the conclusions are well-supported by the data. Overall, I am highly enthusiastic about the potential of this approach.

One limitation of this study is that it did not directly address whether binding of a tagged type I or type II receptor in the context of a ligand-bound receptor heterotetramer might affect the binding of the other receptor type. In my mind, it is critical to address this issue if this tagging strategy is to reach its full potential. In this regard, I think it's important to note that the 33 kDa HALO and 20 kDa SNAP tags are larger than any of the TGF- β family receptor ECDs (largest is ca. 14 kDa). In addition, the TGF- β family ligand dimers are typically 25 kDa and there are cases where the distance separating the N-terminal of one receptor from a nearby bound receptor of the other type is not that great. In the TGF- β :T β RII:Alk5 complex, for example, the N-terminus of T β RII is in direct contact with Alk5 and the N-terminus of Alk5 is only about 23 Å from T β RII (and 25 Å from TGF- β). In the structure of the BMP-2:ActRIIB:Alk3 complex, the N-terminus of Alk3 is only about 20 Å from ActRIIB (and 16 Å from BMP-2).

In the data shown in Fig. 3, the authors suggest that the higher affinity of T β RII to bind TGF- β compared to Alk5, enables detection of robust binding with fluorescently labeled TGF- β 1 with T β RII-Halo, but not Alk5-Halo. However, in the case of Alk5-Halo binding, this assumes there is no T β RII present (and thus Alk5 binding is weak). However, the reality is that these cells almost certainly express T β RII, and thus a robust interaction should be observed between fluorescently labeled TGF- β 1 bound to endogenous T β RII and Halo-Alk5 (since Alk5 actually binds with higher affinity to the TGF- β 1:T β RII complex than T β RII binds to TGF- β 1; see Radaev, et. al, 2010). In light of this, I think a more likely interpretation for the lack of Alk5-Halo binding to fluorescently labeled TGF- β 1 is interference due to the 33 kDa Halo tag on Alk5.

I think too that the unexplicable binding of ActRIIB-Halo to BMP-9 could be explained by preferential higher affinity binding of BMP-9 to endogenous Alk1 and the inability of ActRIIB-Halo to bind due to interference with Alk1 by the Halo tag on ActRIIB.

In summary, in several cases with high affinity receptors, it is clear that Halo- and SNAP-tagging of the N-termini does not impair binding – however, whether it will work in all cases, particularly for type

I and type II receptors in the context of a crowded ligand-receptor heterotetrameric complex, is not at all clear.

Some other more technical concerns are as follows:

1. On line 60, since the type I receptors bind across the dimer interface, but the type II receptors do not, I believe this should read upon "upon type I receptor binding", not "upon type II receptor binding".

2. On line 103, this statement generalizes and is not correct for TGF- β s (TGF- β bind their type II receptor through the fingertips).

3. Some abbreviations should be spelled out, for example STED.

4. Line 125: full library of type I and type II receptors is inaccurate as AMHRII was not studied.

5. Recently, the structure of BMPRII bound to ActB was reported (<https://pubmed.ncbi.nlm.nih.gov/35643319/>); this may enable construction of a more accurate model of the BMPRII:ActA complex.

6. It would be helpful if the authors could comment on the extent to which internalization of ligand-receptor complexes might affect the observed results.

β

Dear Reviewers,

thank you very much for considering our manuscript for Communications Biology. We have followed all comments/suggestions of all three reviewers and are convinced, that our revised manuscript can now be accepted for publication. Thank you for giving us the time to add new experiments and to thereby strengthen our findings.

Below you will find the point-by-point responses to each of the reviewer's comments.

On behalf of my coauthors, we appreciate your time and efforts.

With kind regards,
Petra Knaus

Response letter- color code

1. Reviewer point in italic

Comment of authors in blue

Reviewer #1 (Remarks to the Author):

The manuscript by Jatzlau et al describes the design and production of a library of TGFβ type 1 and type 2 serine/threonine kinase receptors with N-terminal Halo- and SNAP-tags. These chimeric receptors are able to self-label with the addition of specific chemicals linked to fluorophores, providing a platform for measuring co-localization of the ligand with the receptor. The authors proceed to utilize the library to measure binary interactions of labeled TGFβ ligands with the receptors and perform specificity experiments where modifications to the receptors are analyzed for altered ligand specificity.

Overall, this paper functions largely as a methodology paper, describing a specific experimental tool and providing some examples of how this system might be used as a form of validation. The tool itself is novel, filling a gap in the field's ability to explore ligand-receptor interactions in a living cell. This is in part, due to the lack of antibodies that specifically bind to each receptor that do not interfere with ligand binding. Accordingly, it might result in important advances in the understanding of growth factor-receptor interactions across the TGFβ signaling pathway (as hinted in the final paragraph of the discussion). With this in mind, the manuscript falls short of advancing the field in a major direction.

1. Can the platform be used to measure both type I and type II receptors at the same time or is it only useful for high affinity interactions? Collectively, this leaves the reader wondering what are the capabilities and limitations of the system. Thus, significant modification would be needed to improve the manuscript

We completely agree with the referee that current limitations of the assay should be highlighted. We therefore added a new paragraph to the discussion (line 387-409), which address these limitations and future perspectives. Further, we include here a newly performed LSBA experiment, which addresses Activin A-Cy5 binding to COS-7 cells, transiently co-expressing different Snap- and HALO-tagged type I (ALK2, ALK4) with type II receptors (BMPR2, ACVR2A, ACVR2B) (**Response Letter Fig. 1**). As shown in **Fig. 2** of the manuscript, Activin A binding is dependent on co-expressed type II receptor, as indicated by a positive Pearson r-value.

Single expression of either ALK2 or ALK4 did not show a strong correlation to increased Activin A binding. However, while ALK4 co-expression did not alter type II dependent Activin A binding compared to single type II receptor expressing cells, ALK2 co-expression negatively influenced ACVR2A-dependent Activin A binding. In contrast ACVR2B and ALK2 co-expression strongly correlated with Activin A binding. These results indicate that the platform is suitable to study ligand binding in the presence of modulators such as type I and type II receptors or co-receptors. As this manuscript is focusing on the microscopic analysis of single receptor:ligand binding properties, we currently

plan to study ligand binding towards receptor complexes in a consecutive manuscript, for which we will have to develop additional tools.

Response Letter Figure 1. Activin A-Cy5 binding by BMP type I / type II receptors: COS-7 cells were seeded on coverslips and transfected with indicated Halo-tagged type II receptor constructs. 24 hours post transfection, cells were incubated with non-permeable fluorescent Halo-tag substrate CA-AlexaFlour488, SNAP-tag substrate BG-Surface549 and Activin-Cy5 for 30 minutes at 4 °C, fixated with methanol for 5 minutes at room temperature and mounted on glass slides. Cells were imaged at a confocal microscope and 10 cells per condition and replicate were analysed with a semi-automated Fiji ImageJ macro pipeline for assessment of fluorescent growth factor binding (Activin A-Cy5) and fluorescence intensity of receptors. Four ROIs of 100 μm^2 were quantified in each cell. Data shown is derived from 3 independent experiments. Activin A-Cy5 surface binding is represented as relative fluorescence intensity per area. Data is normalized to the highest value per experiment. Linear regression and correlation analysis of ligand:receptor binding based on Cy5-fluorescence intensity and normalized receptor fluorescence (CA-Alexa488 or BG-Surface549) per area. Pearson r value and p-value are indicated. (CA: chloroalkane, BG: benzylguanine).

2. The results are highly dependent on consistent expression of receptors.

Truncated gel analysis makes it difficult to interpret if there are any aggregation of receptors, or engineered receptors.

We performed all Western blot analysis under denaturing conditions and have never observed any bands indicative of receptor aggregates with a higher molecular weight. This can also be seen in the newly added **Supp. Fig. 9F** which shows that both HALO-tagged type I and type II receptors run at the expected molecular weight, without any additional signals being detected. Exemplary untruncated western blots for Supp. Fig. 9F indicate no additional signals (Response Letter, Fig. 2).

Response Letter. Figure 2. Untruncated western blot analysis of HEK293t cells expressing Halo-tagged type I and type II receptors as shown in Supp. Fig. 9F: HEK293t cells were seeded and transfected the next day with Halo-tagged type I and type II receptor constructs (as indicated in Supp. Fig. 9F). 24 hours post transfection, cells were lysed in 1x Laemmli and lysates were subjected to western blot analysis using a specific antibody directed against the Halo-tag. Red frames indicate data shown in Supp. Fig. 9F.

3. Since the receptors have been modified with certain tags that can be recognized via FACS, I would request the authors analyze and normalize all results with a more quantitative measurement such as FACS toward either tag.

We thank the reviewer for the suggestion to add additional control experiments to strengthen the point of the manuscript. As requested, we repeated the analysis of Activin A-Cy5 binding towards cells expressing different HALO-tagged type II receptors followed by FACS measurements. We added these results to the manuscript in the **new Supp. Fig. 5** and described the results in

line 157-162. Indeed, FACS analysis allows for measuring ligand binding to cells transiently expressing high-affinity receptors. We confirmed that Activin A has the highest affinity to ACVR2B > ACVR2A > BMPR2, as shown by the LSBA (**Fig. 2**). However, the increase in Activin A binding was much lower in FACS compared to LSBA (e.g. BMPR2: LSBA ~27 fold; FACS~2.8 fold) likely due to the different experimental approach, indicating that the LSBA is much more sensitive to measure small differences. Further there was no endogenous Activin A binding detected using FACS, while with LSBA we have detected low levels of endogenous Activin A binding (Fig.2C). We addressed the advantages and limitations of both methods in the discussion (lines 387-409).

Supplementary Figure 5: Supplement to Fig.2, (A-C) COS-7 cells were seeded in 6-wells and transfected with indicated Halo-tagged type II receptor constructs. 24 hours post transfection, cells were incubated with fluorescent Halo-tag substrate CA-Alexa488 (green) and Activin-Cy5 (magenta) for 30 minutes at 4 °C and detached with Accutase and analysed using FACS. **(A)** FACS plots show the side and forward scatter (area), count/FITC-A plot, and the gates used to analyse transfected AF488⁺ cells only. Transfected cells were gated for 1000 AF488⁺ cells in gate P2. **(B)** Activin A-Cy5 surface binding is represented as fluorescence intensity relative to unstimulated (w/o) TGFR2 control. Data is shown as F.I. ± SD. Significance was calculated using two-way ANOVA and Dunnett's post-hoc test. **** p < 0.0001 ≡ significant relative to Activin A-Cy5 stimulated TGFR2-expressing cells. **(C)** Plots show AF488⁺ versus Cy5 fluorescence. Simple gating by quadrants allowed to depict the receptor-expression dependent increase in Activin A-Cy5 binding for ACVR2A, ACVR2B and BMPR2 (Q2), whereas untransfected or TGFR2 transfected cells show no detectable Activin A binding. Figure is representative of three different experiments.

4. The manuscript was prepared by multiple primary authors. While this is frequently the case, here noticeable differences between quality of writing, experimental thoroughness, and rigor of reporting between sections are apparent. This is particularly noticeable when comparing section 2 and section 3 of the results section, and the corresponding methodology. The preparation and labeling of BMP9 and TGF β 1 are thoroughly described, while this is less true for Activin A.

We apologize and have added missing details on the preparation of fluorescently labeled recombinant Activin A in the methods section (line 489-506) and adapted a common language style.

5. There was some experimental validation that Halo-tagged receptors could still function in a separate experimental system (BRE luciferase reporter assay) for ALK1 and ALK2. However, this was not performed for ACVR2B.

We added a **new supplemental Figure S9**, which shows that HALO-tagging does not influence ligand:receptor binding and signaling. In Supp. Fig. 9 C-E we compared expression of HALO- or HA-tagged receptors from each ligand class, stimulated with respective ligands and tested signaling competence using BRE₂ or CAGA₁₂-luc in HEK cells. Similar to HA-tagged ALK4 or ACVR2B, HALO-tagged ALK4 and ACVR2B do not increase Activin A signaling compared to ROR2 control, however increased basal signaling is increased when combining both HALO-tagged ALK4 & ACVR2B as well as HA-tagged ALK4 & ACVR2B (Supp. Fig.9C). In contrast, ALK1 and TGFBR2 expression does lead to an increase in BRE₂ or CAGA₁₂-luc response respectively, independent of the tag used (Supp. Fig.9D/E). Collectively, this highlights together with newly added homology modelled ligand:HALO-receptor complexes (Supp. Fig.9A/B) that HALO-tagging does not interfere with ligand binding of BMP/TGF β receptors.

Supplementary Figure 9: Supplement to Fig.3 (A-B) Homology modelling of different Halo-tagged receptor complexes to evaluate the sterically hinderance exerted by the Halo-tag on tetrameric receptor formation. Flexible, long Gly5 linkers allow the formation of tetrameric receptor complex including halo-tagged receptors. **(A)** Tagged type I receptors, from left to right: GDF11:ALK4-Halo:ACVR2B (derived from PDB 7MRZ,⁶) BMP9:ALK1-Halo:ACVR2B (4FAO,¹²) TGFβ1:ALK5-Halo:TGFBR2 (derived from PDB 3KFD,³⁰) **(B)** Tagged type II receptors (left to right): GDF11:ALK4:ACVR2B-Halo (derived from PDB 7MRZ,⁶) BMP9:ALK1:ACVR2B-Halo (derived from PDB 4FAO,¹²) TGFβ1:ALK5:TGFBR2-Halo (derived from PDB 3KFD,³⁰). Halo-fused-receptor models were generated using the PDB entry 5UXZ⁶⁶ for the Halo-tag structure and Rosetta Commons modelling suite after preparation in PyMol. **(C-E)** After one day of transfection with **(D)** the SMAD1/5/8-sensitive (BRE)₂-luciferase reporter or **(E)** the SMAD2/3-sensitive (CAGA)₁₂-luciferase reporter and ROR2-HA in combination with indicated respective **(C)** Activin A-, **(D)** BMP9- or **(E)** TGFβ1 receptors. HEK293t cells were starved for 5 h and stimulated with Activin A (2 nM), BMP9 (0.3 nM) or TGFβ1 (0.2 nM) overnight. Relative Luminescence Units (RLU) are expressed as mean fold induction ±SD over unstimulated ROR2-HA.

HA transfected control cells (n=3 independent experiments). Statistical significance relative to unstimulated ROR2-HA was calculated using two-way ANOVA and Tukey's post-hoc test.

6. *More consistency between sections would improve the manuscript.*
 - *Overall, the validation experiments of this system were lacking. This is of particular importance when trying to quantify what is essentially a qualitative assay system, like microscopy.*

As indicated in the points above new control experiments, i.e. dual luciferase assays and homology modellings, have been added to validate our findings and to also exclude functional interference of the used tags.

7. *While I appreciate the BRE luciferase confirmation that the tagged receptors can induce signaling, it would have been nice to compare these results to WT receptor (no tag).*

As described in Reviewer 1 Point 5 above, we compared the signaling capacity of HALO-tagged receptors with HA-tagged receptors using the dual luciferase assay (see **new supplemental Figure S9**). We did not observe significant differences between the respective receptors. We used HA-tagged receptors (9 amino acid tag at the same location), instead of no tag receptors as these have been used in numerous independent studies and accepted to not interfere with the signaling capacity. In addition, the expression control using anti-HA in each sample is better suited to compare expression levels as individual antibodies.

8. *An alternative would have been to compare ligand binding affinities between WT and tagged receptors using orthogonal/biophysical approaches. While the relative mean fluorescent readings did overall agree with the reported literature (ActA binds more strongly to ACVR2B in both this system and in literature reports of SPR), a more direct comparison would have been appreciated. In particular, this could have answered some of the discrepancies the authors observed in areas where their observations did not match up with the literature, such as the interaction between BMP9 and ACVR2B (Results Section 3, Fig. S7).*

We agree with the reviewer on this point, however the focus of this method was to measure ligand binding on whole cells, where the receptors are exposed to the ligands at their natural environment, i.e. the plasma membrane. Orthogonal biophysical approaches are of course interesting to fill the gap for the discrepancies and certainly be important for future studies.

9. *In section 6, human BMP9 binds to ALK1 from different teleost species that do not resemble human ALK1 across any of the residues that were deemed important*

to binding in section 5. Does this finding discount the findings in section 5? Does this imply that there are multiple possible binding sites/orientations? This point is mentioned in the text of the manuscript, but could the authors provide a possible explanation?.

This point is very well taken: in our analyses human BMP9 showed strong binding to human ALK1, weaker to medaka Alk1 and no binding to zebrafish Alk1. This reduced/loss in binding to the fish receptors can be explained by the lack of the F2 loop in both medaka and zebrafish Alk1 and more precisely the differences in the pre-helix loop in zebrafish Alk1. Interestingly, when transiently expressed in HEK cells and tested for their signaling capacity (human BMP9 in the BRE₂-luc assay), all three ALK1 species led to an increased responsiveness to BMP9. This suggests that even though surface sequestration of BMP9 by medaka and zebrafish Alk1 is less efficient compared to human ALK1, a functional signaling complex might still be stabilized in the presence of the endogenous type II receptors. This we explained in the revised manuscript lines 284-287.

10. Figure 5 E is not all that convincing. The error bars for the different version of Alk2 that show binding are very large. The background is increasing considerably. This data does not seem strong enough to support that different version of the swaps of Alk1 into Alk2 are responding to BMP9 treatment. More rigorous analysis would be better, including titrations.

The experiment has been repeated including a non-binding receptor as control (ROR2-HA). The increased BMP9 responsiveness of cells expressing V2 & V3 ALK2^{MimicALK1} is clearly visible. The graphic was replaced in Figure 5 (see **NEW figure 5**).

11. The Western-blot validation of construct expression levels is inconsistent. *ALK7* never really seems to express well (Fig. S1). Of more concern, *ALK2-Halo* does not seem to express at similar levels to all other receptors in Fig. S9, calling into question whether the conclusions of section 5 are even applicable. Maybe there was little binding of BMP9 to the cells transfected with *ALK2-Halo* because the transfection didn't work. Maybe the reason that there was less binding of BMP9 to *V3 ALK2Mimic-ALK1-Halo* is that less of it expressed than of the other *ALK2* mutants. Again, these results should somehow be normalized, perhaps thorough FACS.

Western-blot validation for *ALK2^{MimicALK1}* has been repeated and shows equal expression among the different constructs. The before observed discrepancy in expression levels of *ALK2^{MimicALK1}* was likely attributed to insufficient DNA quality for one of the constructs. The figure has been replaced (see **New Supp. Fig. 12A**).

In regard to the normalization of the receptor/ligand intensity values, this analysis has always been included and is seen in any of the linear regression diagrams throughout the manuscript (e.g. Fig. 3D). Here we analyzed if increased receptor expression (higher HALO-bound CA-Alexa488 intensity) would also lead to higher ligand binding and hence ligand intensity. However, in contrast to *ALK1*, *ALK2* expression does not correlate with increased BMP9 surface binding.

12. For consistency, Western-blot analysis of receptor transfection into HEK293T cells, in addition to COS-7, would have helped inform the results of the BRE luciferase experiments.

Western-blot analysis has been added for HEK293T cells and added to the manuscript in **New Supp. Fig. 9F** and **New Supp Fig.12C** (see also below) and highlights equal expression levels for all constructs used in the reporter gene experiments.

13. What is the endogenous expression of these receptors in COS-7 cells? What effect, if any, might this have on the overexpression of these tagged receptors?

RNA sequencing data of COS7 and U2OS cells has been added (**New Supp. Fig. S3D,E.**) Since the expression of overexpressed receptors is under the control of the CMV promoter it is unlikely that the level of endogenous receptors influences the expression levels of the plasmid-based receptor. A contribution of the endogenous receptors on ligand binding to the overexpressed receptors is possible in the LSBA. However, the amount of endogenous expressed is likely not sufficient to impact on ligand binding to the overexpressed receptor. In order to highlight that the assay gives similar results in different cell types, we repeated the Activin A LSBA in U2OS cells (**new Supp. Fig. 3A-C**) and obtained similar results as in COS7 cells. This crucial control experiment was commented in the results section in line 154-156.

14. The fact that BMP9 binding to WT ALK2 is not detected (as described in Line 305-322) is attributed to relatively lower affinity between BMP9 and ALK2 as compared to ALK1. This implies that this system is only useful for detecting high affinity ligand:receptor interactions, limits the utility of this system (i.e. what are the limitations of the system?). It would have been interesting to determine whether increasing concentrations of labeled BMP9 might have been detectable.

We fully agree on this comment and have also addressed the limitations in lines 384ff. At the same time, we emphasized the advantage of the system as here ligand binding to full length, membrane inserted receptors in living cells can be measured and compared to their signaling capacity in the same, i.e. directly comparable setting. This has been made possible here now for the first time. The suggested increased BMP9 concentration experiment has on purpose not been followed, as we emphasize here the settings under physiological ligand concentrations of 0.3nM.

Minor comments

15. There is a lack of consistency in verb tense used throughout this document. For example: "We here aim..." (Line 104) vs. "Here, we aimed..." (Line 133). This is present throughout the Introduction and Results sections and should be corrected.

The verb tense was adapted to past tense in all cases.

16. *Reword line 144-147 as correlation is not causation. (i.e. not leads to more ligand binding)*

Sentence was changed to: " ..., highlighting that increased high-affinity receptor expression allows more ligand binding" (Line 153-154)

17. *Change "...dyes allow to exclusively study..." to "dyes that allow the exclusive study of..." (Line 126).*

Line has been changed as proposed by the reviewer (Line 127).

18. *Line 197 links to Fig. S6C. There is no Fig. S6C, only A and B.*

Due to the changes proposed by reviewer 3 regarding a different BMPR2 ECD, we excluded this sentence and the reference. See reviewer 3 comment 8.

19. *In Fig. 6D, the graph is missing the bars for untreated cells transfected with the different ALK1 paralogues.*

The missing data has been added to the diagram and the figure has been replaced in Fig. 6D.

20. *In Fig. S4, it would also be nice to have this data for ActA-Cy5.*

Fig. S4 depicts the synthesis of NHS-SiR-d12, NHS-activated Cy5 was commercially obtained (Lumiprobe, cat. no 13020), which is why we can't provide the corresponding figure for the synthesis of NHS-Cy5. More details on the labeling of Activin A-Cy5 have been added to the method section as described above.

Reviewer #2 (Remarks to the Author):

In this manuscript Jatzlau and co-workers describe an elegant assay making use of differentially tagged receptors of the TGF β /BMP superfamily in combination with fluorescently labelled ligands for optical quantification of receptor-dependent ligand binding. This is a well designed and well conducted study providing a valuable assay to evolve our understanding of the TGF β superfamily and its down stream effectors. There are some issues however that need further clarification.

- 1. In a family of signaling molecules where expression levels and ratios of receptors will influence function and outcome, how does overexpression influence the outcome?*

In our experimental approach overexpression allows to assess binding predominantly to the overexpressed receptor, i.e. above background of endogenous receptors. All data is normalized and referenced towards controls, which resemble only endogenous receptors levels. To address the point of different endogenous receptor levels, we now added new control experiments using also U2OS cells as discussed in reviewer 1 point 13 and below (see **new figures Suppl Fig 3A-E**).

- 2. Most of the studies have been performed in COS-7 cells expressing certain members of the TGF β family on their cell surface. Would the outcome be different when different cell types were used?*

We appreciate the question of the reviewer and therefor repeated as mentioned above the LSBA using Activin A in U2OS cells, which gave the same result as observed for COS-7 cells (**New Supp. Fig. 3A-C**), highlighting that ligand binding is primarily dependent on transiently expressed high-affinity receptors. This information has been added to the manuscript in Line 154ff and to **New Supp. Figure 3A-C** thereby increasing the value of the manuscript (see below). We do not exclude that ligand binding of some members of the TGF β superfamily might vary in dependence of endogenous receptor or co-receptor (e.g. TGF β 2/TGFBR3) expression.

Supplementary Figure 3: Supplement to Fig.2 (A-C) Transiently transfected U2-OS cells expressing ACVR2A-, ACVR2B-, BMPR2-, TGFR2- or ALK4-Halo were 24 hours post transfection simultaneously incubated with Halo-tag substrate CA-Alexa488 (green) and Activin A-Cy5 (magenta). Data shown is derived from 3 independent experiments. **(A)** Representative confocal microscopy images of Activin A-Cy5 stimulated U2-OS cells expressing type II receptors (ACVR2A, ACVR2B, BMPR2, TGFR2) or type I receptor ALK4. Scale bar $\cong 20 \mu\text{m}$. **(B)** Activin A-Cy5 surface binding represented as fluorescence intensity per area relative to untransfected cells. Data is shown as F.I. \pm SD. Significance was calculated

using two-way ANOVA and Tukey's post-hoc test. **** $p < 0.0001$ \equiv significant relative to all stimulated conditions. **(C)** Linear regression and correlation analysis of ligand:receptor binding based on normalized Cy5-fluorescence intensity and normalized receptor fluorescence (CA-Alexa488) per area ($n = 3$). **(D-E)** Endogenous BMP & TGF β receptor expression in COS-7 (GSM5461004) and U2-OS cells (GSM4255919) extracted from publicly available RNA-Seq. data (GEO Data base)^{64,65}.

3. *As TGFbeta type 3 receptors modulate affinity as well, would it be possible to include them in the assay?*

In future studies, we are planning to address and optimize ligand binding studies in dependence of double-receptor expression or co-receptor expression. In order to study the effect of ligand binding to TGFBR3, Halo or SNAP tagging should be C-terminal as N-terminal tagging of the ECD might interfere with the binding mode of TGFBR3 towards TGF β 2 (Villarreal et al. 2016; doi: 10.1021/acs.biochem.6b00566), while N-terminal Halo-tagging of TGFBR2 did not interfere with TGF β 1 binding (**Fig. 3, New Supp. Fig.9B**). During the revision we added data showing the potential to investigate ligand binding towards type I and type II expressing cells. However, expression levels of single receptor expressing cells vary compared to double transfected cells, making a direct comparison more complicated (Reviewer 1 Point 1; Rebuttal Figure 1). We are currently working on a way to quantify receptor complex formation using the Halo- and SNAP-BMPR library. We address this point in a new section of the discussion covering limitations and future perspectives of the method in line 384ff.

4. *TGFb ligands bind a dimers to the receptor complex. Is the labelled ligand monomeric or dimeric?*

We added the information that all ligands used for labelling and ligand-affinity measurements are dimeric in the result and method section.

5. *Some of the effects are related to heterodimeric proteins composed of e.g. BMP9/BMP10. Can the authors speculated if this can be included in the assay our would make interpretation and translation more difficult.*

This is a great question. A requirement would be a sufficient production of recombinant heterodimeric BMP9/BMP10, allowing NHS-fluorophore labeling and purification. Alternatively, labeling can be optimized using site-directed incorporation of unnatural amino acids, which would allow targeted labeling, or even dual color labeling of a heterodimeric ligand. Subsequently, a heterodimeric ligand could equally be used for surface ligand binding studies and be directly compared to homodimeric ligands.

The advantage of the system is that besides ligand binding, subcellular information as well as dynamic turnovers can be investigated. This point has been added to the discussion in line 404ff as a future perspective for the assay.

Reviewer #3 (Remarks to the Author):

The current understanding of ligand-receptor binding and assembly in the TGF- β family has mainly been achieved from structural studies and SPR-based binding studies with purified ligands and receptor ECDs. The new approach proposed by Jatzlau and co-workers, which uses HALO- and SNAP- N-terminally tagged type I and type II receptors, extends this by allowing one to study assembly as it occurs on the surface of live cells. Thus, with this approach, it should be possible to study all ligand-receptor interactions occurring simultaneously with a ligand, rather than just one at a time. This is important as this can extend understanding to include effects arising from membrane localization, receptor competition, co-receptor binding, among other possible effects.

The results presented focused on studies of some singly-tagged receptors interacting with a given target ligand. The tagged receptors are all clearly synthesized and reach the cell surface and imaging of these, with fluorescently labeled ligands, nicely recapitulates previous binding measurements for ActA, TGF- β 1, and BMP-9. The measurements also appear to be sufficiently sensitive to detect relatively modest gains of affinity, as demonstrated through structure-based engineering of BMPRII and Alk2. The technical quality of the work is outstanding, and the conclusions are well-supported by the data. Overall, I am highly enthusiastic about the potential of this approach.

One limitation of this study is that it did not directly address whether binding of a tagged type I or type II receptor in the context of a ligand-bound receptor heterotetramer might affect the binding of the other receptor type. In my mind, it is critical to address this issue if this tagging strategy is to reach its full potential.

- 1. In this regard, I think its important to note that the 33 kDa HALO and 20 kDa SNAP tags are larger than any of the TGF- β family receptor ECDs (largest is ca. 14 kDa). In addition, the TGF- β family ligand dimers are typically 25 kDa and there are cases where the distance separating the N-terminal of one receptor from a nearby bound receptor of the other type is not that great. In the TGF- β :T β RII:Alk5 complex, for example, the N-terminus of T β RII is in direct contact with Alk5 and the N-terminus of Alk5 is only about 23 Å from T β RII (and 25 Å from TGF- β). In the structure of the BMP-2:ActRIIB:Alk3 complex, the N-terminus of Alk3 is only about 20Å from ActRIIB (and 16 Å from BMP-2). In the data shown in Fig. 3, the authors suggest that the higher affinity of T β RII to bind TGF- β compared to Alk5, enables detection of robust binding with fluorescently labeled TGF- β 1 with T β RII-Halo, but not Alk5-Halo. However, in the case of Alk5-Halo binding, this assumes there is no T β RII present (and thus Alk5 binding is weak). However, the reality is that these cells almost certainly express T β RII, and thus a robust interaction should be observed between*

fluorescently labeled TGF- β 1 bound to endogenous T β RII and Halo-Alk5 (since Alk5 actually binds with higher affinity to the TGF- β 1:T β RII complex than T β RII binds to TGF- β 1; see Radaev, et. al, 2010). In light of this, I think a more likely interpretation for the lack of Alk5-Halo binding to fluorescently labeled TGF- β 1 is interference due to the 33 kDa Halo tag on Alk5.

We absolutely agree that one expects endogenous binding of fluorescently labeled TGF- β 1 as COS-7 cells express both ALK5 and TGFBR2 (New Supp. Fig.3D). As expected, we see an increase of TGF- β 1-SiRd12 binding to untransfected COS-7 cells (Fig. 3C). However, we do not observe an increase of TGF- β 1 binding with ALK5 overexpression, as we normalize to the endogenous binding level. It is likely that most of endogenous TGFBR2 is already engaged in ALK5:TGFBR2:TGF- β 1 complexes and none would be free for complexes with the overexpressed ALK5-HALO. That is why TGF- β 1 binding levels are equal to the untransfected control.

In order to proof that N-terminal HALO-tagging does not impede with ALK5 and TGFBR2 ligand binding we performed homology modelling of Halo-tagged receptor complexes to evaluate the sterically hinderance exerted by the HALO-tag on tetrameric receptor formation. We therefor used PDB: 3KFD (Radaev, et. al, 2010) and modelled TGFBR1:ALK5-HALO:TGFBR2 and TGFBR1:ALK5:TGFBR2-HALO (Supp.Fig. 9 A-B). As shown in the **newly added figure**, the HALO-tag is connected to the ALK5-ECD with a long flexible linker which should allow it to move freely. The visualization shows just one possible conformation.

Additionally, we tested the signaling competence of ALK5-HALO and TGFBR2-HALO and compared them with HA-tagged variants using the CAGA₁₂-Luc assay (**New Supp. Fig.9E**, see below). In line with the LSBA data, we observe a strong increase in TGF- β 1 response when TGFBR2-HALO or TGFBR2-HA are transiently expressed, but not when ALK5-HALO or ALK5-HA are expressed. However, in line with the reviewers comment, expression of both ALK5 and TGFBR2 further increases already the baseline response, independent of the used tag. This valuable control experiment was added to the manuscript and together with the homology modelling argues against a sterically interference of the HALO-tag.

This figure was added as **NEW Supp. Figure 9E**.

(E) the SMAD2/3-sensitive (CAGA)₁₂-luciferase reporter and RLTK-luc in combination with indicated respective ... (E) TGFβ1 receptors. HEK293t cells were starved for 5 h and stimulated with ... TGFβ1 (0.2 nM) overnight. Relative Luminescence Units (RLU) are expressed as mean fold induction ±SD over unstimulated ROR2-HA transfected control cells (n=3 independent experiments). Statistical significance relative to unstimulated ROR2-HA was calculated using two-way ANOVA and Tukey's post-hoc test.

2. I think too that the unexplicable binding of ActRIIB-Halo to BMP-9 could be explained by preferential higher affinity binding of BMP-9 to endogenous Alk1 and the inability of ActRIIB-Halo to bind due to interference with Alk1 by the Halo tag on ActRIIB.

As mentioned above, we also addressed this issue using homology modelling of BMP9:ALK1:ACVR2B-Halo (derived from PDB 4FAO, see below) and would expect no sterical hinderance of ACVR2B/BMP9 binding by the flexible HALO-tag. This is in line with successful binding of Activin A to the same HALO-tagged ACVR2B (Fig. 2). Further, a contribution of endogenous ALK1 to BMP9 binding in COS7 cells is unlikely as ALK1 is not expressed in these cells (**New Supp. Fig. 3D, see below**), but expressed specifically in endothelial cells. In our analysis we did focus at the moment solely at the binding potential of single receptors. As mentioned above (reviewer 1 comment 1), we are currently in the process of developing the LSBA further to address a co-dependence of dual-receptor binding.

3. *In summary, in several cases with high affinity receptors, it is clear that Halo- and SNAP-tagging of the N-termini does not impair binding – however, whether it will work in all cases, particularly for type I and type II receptors in the context of a crowded ligand-receptor heterotetrameric complex, is not at all clear.*

We completely agree with the reviewer that potential oligomerization states of clustered receptors could be affected. However to strengthen the point that N-terminal tagging does not interfere with the ligand:receptor interface, we added as mentioned above in the **new Supp. Fig. 9A-B** (see below) homology models of different Halo-tagged receptor complexes to evaluate the sterically hinderance exerted by the HALO-tag on tetrameric receptor formation. Using a flexible Glycine 5 linker, the HALO-tag has enough freedom to not interfere with ligand:receptor binding. This has been confirmed by the data presented in this manuscript. However, for certain receptor-ligand binding modes (e.g., TGFBR3 ligand binding; see Reviewer 2 comment 3) HALO-tagging should be C-terminal as N-terminal tagging of the ECD might interfere with ligand binding.

Supplementary Figure 9: Supplement to Fig.3 (A-B) Homology modelling of different Halo-tagged receptor complexes to evaluate the steric hindrance exerted by the Halo-tag on tetrameric receptor formation. Flexible, long Gly5 linkers allow the formation of tetrameric receptor complex including halo-tagged receptors. **(A)** Tagged type I receptors, from left to right: GDF11:ALK4-Halo:ACVR2B (derived from PDB 7MRZ,⁶) BMP9:ALK1-Halo:ACVR2B (4FAO,¹²) TGFβ1:ALK5-Halo:TGFBR2 (derived from PDB 3KFD,³⁰) **(B)** Tagged type II receptors (left to right): GDF11:ALK4:ACVR2B-Halo (derived from PDB 7MRZ,⁶) BMP9:ALK1:ACVR2B-Halo (derived from PDB 4FAO,¹²) TGFβ1:ALK5:TGFBR2-Halo (derived from PDB 3KFD,³⁰). Halo-fused-receptor models were generated using the PDB entry 5UXZ⁶⁷ for the Halo-tag structure and Rosetta Commons modelling suite after preparation in PyMol.

4. *Some other more technical concerns are as follows: On line 60, since the type I receptors bind across the dimer interface, but the type II receptors do not, I believe this should read upon “upon type I receptor binding”, not “upon type II receptor binding”.*

The respective sentence was not changed as it refers to the restrained variability of the Activin A dimeric structure upon type II receptor (high affinity) binding, which subsequently allows type I receptor recruitment and binding.

5. *On line 103, this statement generalizes and is not correct for TGF-βs (TGF-β bind their type II receptor through the fingertips).*

We changed the sentence to specify our statement to BMPs (line 104).

6. *Some abbreviations should be spelled out, for example STED.*

STED is now spelled-out in the results and method section.

7. *Line 125: full library of type I and type II receptors is inaccurate as AMHRII was not studied.*

Full library was exchanged for “a comprehensive library”.

8. *Recently, the structure of BMPRII bound to ActB was reported (<https://pubmed.ncbi.nlm.nih.gov/35643319/>); this may enable construction of a more accurate model of the BMPRII:ActA complex.*

We thank the reviewer a lot for this suggestion, we were aware that during the process of revision the structures became available (PDB 7U50). We agree that a BMPR2 structure bound to Activin B is a better starting point than in unbound state to model the interaction of BMPR2 to Activin A. We decided to re-do the docking prediction on Activin A and the generation of the mutant BMPR2^{-mimic-ACVR2B}. We exchanged the predictions in the manuscript as slight differences in the interface and lower Rosetta scores suggest that we were now able to find an optimized energy minimum. We needed to include another loop modeling step before performing the dockings as BMPR2 was not completely resolved in the Activin B bound structure (PDB 7U50). In Figure 4E, there are now more hints on why ACVR2B binds better to Activin A than BMPR2. For example, I88 in BMPR2 can possibly interact with R87 and I109 of Activin A, but this interaction is probably less favorable than D80 of ACVR2B to R87 of Activin A. Also, the distances between the residues in ACVR2B and BMPR2^{-mimic-ACVR2B} are now more similar to each other than in the docking performed with the unbound BMPR2 structure (PDB 2HLQ).

One thing we realized when working with PDB 7U50 was that the Finger3 of BMPR2 was not resolved, suggesting it is flexible although it is bound to Activin B. Thus, we now assume that the modeled interactions at this additional binding site are transient and more flexible. Therefore, we decided that showing a distinct structure of this area is not possible using our dockings. We decided to remove this panel in Supp. Fig. 10 (prev. Supp. Fig.7C), and the respective comment in the results part respectively. We still think that, as there are residues present in Finger3 of BMPR2^{-mimic-ACVR2B} and BMPR2, which are different in ACVR2B, that can make contact with Activin A, and might contribute to the binding, thereby being one reason why BMPR2^{-mimic-ACVR2B} binds Activin A better than ACVR2B. It will be intriguing to see future studies follow up on this aspect.

9. *It would be helpful if the authors could comment on the extent to which internalization of ligand-receptor complexes might affect the observed results.*

We thank the reviewer for this important question. All LSBA experiments have been carried out at 4°C to limit diffusion, membrane fluidity and endocytosis. Therefore, little to no internalization was observed in the LSBA experiments. However, If the experiment is performed at 37°C for 30 mins or as a live-cell imaging experiment at 37°C receptor & ligand positive vesicles are detectable. With the combination of the cell-impermeable Halo- or SNAP-tag substrates this allows the study of the surface receptor pool. Using two different non-permeable dyes in two consecutive staining steps would also allow to investigate the amount of receptor that is newly trafficked to the plasma membrane. In a study by us, just published in BMC Biology (*BMC Biol* **20**, 210 (2022). <https://doi.org/10.1186/s12915-022-01396-y>), we successfully used SiR-d12-labeled BMP9 to show that endocytosed BMP9 locates to CAV1-positive vesicles in HUVECs without transient expression of any receptor, highlighting the wide range of applicability of labeled-ligands in combination with Halo-/SNAP-tagged surface receptor labeling or endogenous receptor levels.

REVIEWERS' COMMENTS:

Reviewer #1 (Remarks to the Author):

The authors have made significant improvements to the manuscript that gives the reader more confidence in the interpreting the results. This manuscript should be the stepping stone for further analysis of ligand receptor interactions of the TGFb family.

Reviewer #2 (Remarks to the Author):

the authors adapted the manuscript and added substantial additional information that improved the story and I have no further questions.

Reviewer #3 (Remarks to the Author):

I believe the approach the authors describe for snap- and halo- fluorescently tagging type I and type II receptors of the TGF-b family has significant potential; this is demonstrated by the ability to recapitulate the most well known and studied high affinity ligand-receptor interactions in the family; it is further demonstrated through well informed and executed engineering approaches, which demonstrates the ability to detect more subtle changes in ligand-receptor interactions. It would have been nice to see the method extended to studying ligand-mediated heteromeric assembly, as well as detecting known weak, but functionally significant ligand-receptor interactions, but this study fell short of doing so. Nonetheless, I remain enthusiastic and I believe publication in its current form (which I might add was significantly improved in response to the reviewers comments) will inspire further efforts to exploit the full potential of this approach.

Some minor comments and corrections are as follows:

1. In the opening lines of the first and second paragraph of the Introduction, the TGF-b family is inconsistently described as a family and as a superfamily. My bias is toward family, not superfamily, but whatever is chosen should be done consistently throughout the MS.
2. It is incorrect to state that "both hands offer two "wrist" and two "knuckle" epitopes, which bind to type I and type II receptors, respectively" since this does not apply to all family ligands (TGFb type II receptor does not bind to the knuckle).
3. In the Introduction, it states that BMPRII likely binds to the knuckle epitope. However, it is only unlikely if the authors are somehow skeptical of binding of BMPRII to the knuckle epitope of ActB, as recently reported (PMID: 35643319).
4. On line 166, it should be AcvR2B-Fc, not AcvR2B.
5. On line 170, it should be Fig. S7 & S8, not S5 & S6.
6. On line 169, it should be either _____, or _____ (not neither _____, nor _____) as this is a double negative with the preceding not on line 168.
7. On line 189, it should be Fig. 4B-D, not 4C-D.

8. On line 193, it should be Fig. S11, not Fig. S8.